# Bee-pollination promotes rapid divergent evolution in plants growing in different soils

Thomas Dorey[1,2] & Florian P. Schiestl[1] ✉

Divergent evolution leads to variation among populations and thus promotes diversification. In plants, adaptation to different soils, pollinator guilds, and herbivores is thought to be a key ecological driver of adaptive divergence, but few studies have investigated this process experimentally. Here we use experimental evolution with fast cycling *Brassica rapa* plants to study the impact of soil, pollination, herbivory, and their interactions on divergent evolution in various traits during eight generations of selection. We found significant evolutionary changes in plant phenotypes caused by all three factors and their interactions. In the richer soil type, plants showed higher evolutionary rates, especially with bumblebee-pollination, which led to the evolution of increased attractiveness of plants to bumblebees. Plants that had experienced aphid-herbivory showed lower attractiveness. We found the strongest evolutionary divergence when plants evolved in different soils with bee-pollination rather than hand-pollination, irrespective of herbivory. This "soil-pollinator effect" impacted divergence in diverse suites of traits, for example leaf size, flowering time, flower petal length, some floral volatiles and leaf glucosinolates. We conclude that the interaction between soil and biotic pollination may be an important cause for divergent evolution of plants growing on different soil types, even without a shift in pollinator guilds.

The first step in plant speciation is the establishment of genotypic and phenotypic differences among populations, sometimes called plant races[1]. Although both neutral (i.e. genetic drift) and adaptive evolutionary processes can cause such variations, adaptation driven by selection is thought to be of prime importance for divergent evolution[2]. Factors that cause adaptation to local conditions include abiotic factors such as climate[3], mixed abiotic-biotic factors like soil and soil-born microorganisms[4], as well as biotic interactions such as those between plants, herbivores, and pollinators[5–7]. Whereas shifts in physiological-, root- and overall growth-related traits are thought to be driven primarily by soil adaptation[8], divergence in defense mechanisms is expected to be selected for by variable herbivore communities[9], and floral divergence is thought to be driven by shifts in pollinator species or pollinator guild composition[10,11].

Divergent evolution in response to soil-type is common in plants and is well documented[12]. Plant adaptations to soil may be driven by physical and/or chemical parameters such as water-holding capacity, nutrient content[13,14], salinity[15,16], calcium ions in calcareous soil[14,17], or heavy metal content in serpentine-[18] and other metalliferous soils[19,20]. Soil represents the major source of nutrients for plants[21,22], and selection favors mechanisms that improve resource acquisition and/or utilization[23]. Adaptation to physical properties typically causes alterations of the root system, and re-allocations between above- and below ground biomass[24]. On the other hand, physiological mechanisms to cope with chemical properties like the exclusion or tolerance of high concentration of toxic compounds in plant tissue have also been demonstrated[12,25].

Plants in their natural environment are involved in multiple interspecific biotic interactions, for example with animals that

[1]Department of Systematic and Evolutionary Botany, University of Zürich, Zürich, Switzerland. [2]Present address: Department of Environmental Sciences, University of Basel, Basel, Switzerland. ✉e-mail: florian.schiestl@systbot.uzh.ch

pollinate flowers and/or feed on plant tissue[26,27]. By feeding on plants, herbivores induce strong selection on defense mechanisms in plants that may differ depending on the feeding mode and sensitivity of herbivore taxa to defense compounds[9,28,29]. Pollinators, in contrast, act mutualistically and select for signals and morphological traits ensuring optimal pollen transfer[26,30]. As pollinators differ in both their sensory preferences and morphological features, selection imposed by different pollinators can lead to divergent evolution among plant populations/species[31,32].

Multiple lines of evidence show that different ecological factors such as soil-type and plant-animal interactions influence each other at various levels. First, soil-type strongly impacts the morphology and physiology of plants through phenotypic plasticity, and thereby changes the plants' interactions with pollinators and herbivores[33]. For example, soil nutrients have been shown to impact plant phenology, flower production, flower shape, nectar production, as well as nectar and pollen chemistry[34–36], all with high relevance for pollinator behavior. Plants growing on serpentine soils have been shown to receive less pollinator visitations and less herbivory[37], whereas fertile soils can increase plant attractiveness to herbivores[38]. Hyperaccumulation of toxic metals is known to negatively affect herbivores and may also have an impact on pollinators and plant-fertility related traits[39–41]. Such environmentally-induced changes in plant traits likely change patterns of phenotypic selection caused by the behavior of pollinators and herbivores[33,42], leading to evolutionary changes driven by interactions between soil and animal interactors.

In addition, although different ecological factors may typically select for different plant traits, the evolution of those traits is often not independent because of ecological and/or genetic mechanisms[43–46], potentially resulting in trade-offs for plants[47]. For instance, soil chemistry can modify adaptive responses to both plant-herbivore and plant-pollinator interactions through resource limitation. Soil fertility can determine the evolution of plant growth and defense as predicted by the resource availability hypothesis[48,49]. In poor growing conditions, plants are expected to evolve low growth-rates but high levels of constitutive defense, a strategy driven by the lack of resources necessary to compensate for herbivore damage or produce effective herbivore-induced defenses[48,50–53]. In contrast, where resource availability is higher, plants may invest less in defense but evolve an ability to compensate for or tolerate herbivore damage[54–56]. In addition, traits that evolve in response to herbivores are intricately linked with those that evolve under selection by pollinators and vice versa[57]. For example, evolution of defense or herbivore-induced plasticity can compromise plant attractiveness to pollinators by decreasing the attractiveness of floral signals[58,59], and rewards[60,61], thereby causing a trade-off between pollinator attractiveness and defense against herbivores.

Finally, patterns of macroevolutionary diversification suggest combined effects of soil-type and pollinators on plant speciation in some plant lineages and floristic provinces[8]. Evidence for this is found in closely related plant species being adapted to different pollinators as well as to different soil types[62–65]. The evolution of such combined adaptions can be explained by adaptation to one factor, e.g. soil-type, driving the subsequent adaptation to the other (e.g. pollinator), either because pollinator distribution co-vary with soil-type in a mosaic-like fashion (i.e. the Grant-Stebbins model of pollinator driven speciation[6,10]), or soil-adaptation reinforces differential pollinator attraction upon secondary contact[63]. Both scenarios assume a shift in pollinator guilds as the primary force for divergence in flower-types and the subsequent evolution of pollination ecotypes[32,66]. Other studies, however, suggest that pollination ecotypes can also arise as a byproduct of habitat adaptation, such as to elevation ranges[67–69]. Generally, however, we know little about the combined effects of various ecological factors, such as soil, pollinators and/or herbivores in adaptive diversification, and

their importance in promoting divergent evolution at the onset of the diversification process[70].

A powerful way to study the process of diversification is experimental evolution[71], which has, however, up to now only rarely been applied to plants, and most studies that do, focus on the effects of only one ecological factor[31,61,72–74]. Here we aim to contribute to filling this gap by disentangling the effects of soil, bee-pollination, and aphid-herbivory in shaping divergent evolution, by performing an eight-generation evolution experiment with fast cycling *Brassica rapa* plants as an outcrossing model with short generation time and high standing genetic variation[75,76]. In order to apply as-natural-as-possible ecological parameters in the experiment, we mimicked natural conditions found in populations of a Southern Italian *Brassica* species, *Brassica incana*, that grows on limestone- or tuff soil, the latter with higher nutrient availability than the former, with bumblebees and aphids being major biotic interactors. Nevertheless, the objective of the study was not to investigate evolutionary patterns specific to any plant species, but to provide overall insights and create new testable hypotheses about how soil and animal interactors impact plant divergent evolution. We used a 3-way factorial design with two different soil types (limestone- and tuff soil) that were collected in natural populations of *Brassica incana*, with- and without aphid-herbivory, and with either bumblebee-pollination or hand-pollination (Fig. 1). Hand pollination was included as a control for pollinator-mediated selection, as it represents random pollination without pollinator-mediated selection. Each treatment comprised two replicates with 49 plants each, evolving independently during eight consecutive generations; to reduce maternal effects, two generations without insects were included before plants were phenotyped. The soil used in the experiment was not sterilized, because sterilized soil can hardly be kept sterile during the experiment, with unnatural re-establishment of microbes being the possible consequence. We assessed the evolutionary responses of plants after the experiment by phenotyping plants in a common garden environment with standardized soil. We measured morphological as well as chemical traits, namely flower volatiles collected by head-space sorption and analyzed by gas chromatography, and leaf glucosinolates analyzed by liquid chromatography. These different traits are traditionally thought to evolve in response to different ecological factors such as pollinators (floral scent, flower morphology), herbivores (defense chemistry, i.e. glucosinolates), and soil (vegetative traits such as leaves). We performed various statistical analyzes, such as principle component analysis, linear discriminant function analysis and general linear models, and calculated evolutionary rates in haldanes. We addressed the following questions: (1) How do plant morphology, nectar, floral scent, defense traits, and attractiveness to pollinators evolve in response to different soil types, pollination and herbivory treatments? (2) Do interactions between these factors lead to distinct evolutionary trajectories and which combinations lead to the strongest divergence? For (1) we predicted that low nutrient soil should limit evolutionary response to bee-pollination and aphid-herbivory, as resource limitation is expected to induce strong trade-offs between traits. For (2) we predicted that because the ecological factors likely selected for different traits, more factors should cause more divergence[77], and thus the interaction of both herbivory and pollination lead to the strongest divergence among plant populations on different soil types.

Our results show rapid divergent evolution in many traits driven by soil, pollination, herbivory and their interactions in our model system. We document that divergent evolution in response to different soil types was more pronounced with bee-pollination rather than hand pollination. This shows that pollination, the biotic factor with the strongest fitness-impact in our experiment, interacted with soil differences leading to the evolution of different plant phenotypes without a switch in pollinator quantity or quality. We conclude that divergent evolution driven by different soils can be accelerated by

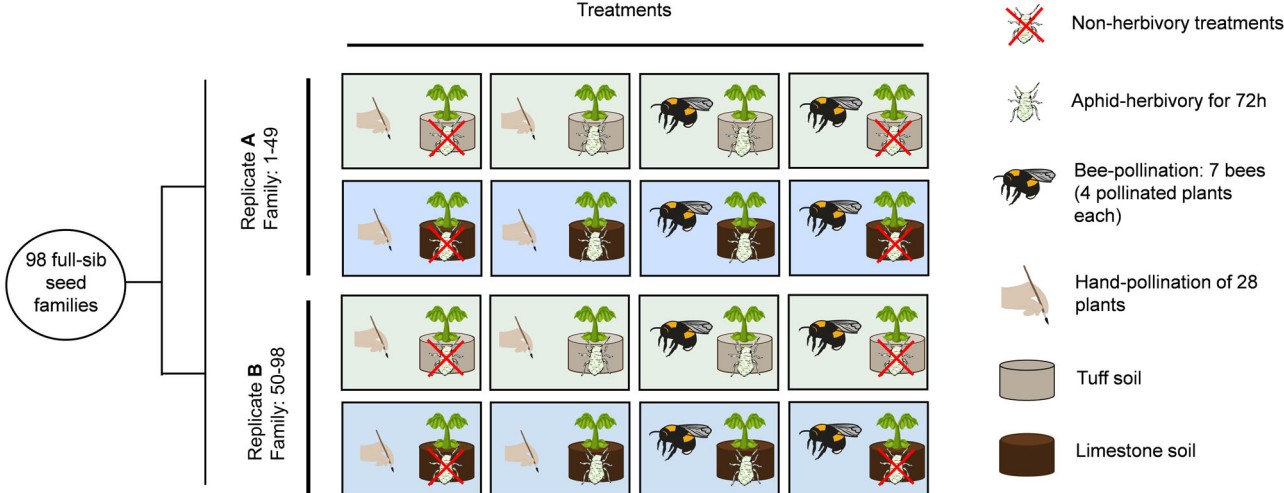

**Fig. 1 | Design of our experimental evolution study.** 98 full-sib seed families of fast cycling *Brassica rapa* plants were divided into two replicates (**A** and **B**), and spread among treatment groups, with each seed family being represented in each treatment (49 plants were used per treatment per replicate). Differently colored squares represent the two types of soil, tuff- (pale green) and limestone soil (pale blue).

biotic pollination, which may act as a starting point for soil-pollinator driven diversification in plants.

## Results

Throughout this Result section, unless explicitly stated otherwise, only statistically significant findings are reported ($P < 0.05$).

### Divergence in individual traits

Soil: The morphology of plants that evolved in tuff during the experiment diverged from plants evolving in limestone soil, with increases in nectar production (1.30-fold), leaf size (1.23-fold), flower number (1.09-fold), plant size (1.08-fold), and flowering time (1.04-fold; Table 1, Supplementary Table 7, Supplementary Data 1). In terms of floral scent, plants that evolved in tuff exhibited distinctive emission profiles compared to those that evolved in limestone, with increases in the emission of benzaldehyde (1.21-fold) and (*E,E*)-α-farnesene (1.13-fold), but decreases in the emission of 2-aminobenzaldehyde (0.83-fold), benzyl nitrile (0.76-fold), phenylethyl alcohol (0.62-fold), phenylacetaldehyde (0.48-fold), and methyl anthranilate (0.38-fold; Table 1, Supplementary Table 7). Plants that evolved in tuff exhibited a higher leaf glucosinolates production (1.12-fold) than those that evolved in limestone. This change was particularly pronounced for glucobrassicanapin (1.38-fold), gluconapin (1.15-fold), and glucoasturtiin (1.15-fold; Fig. 2, Table 1, Supplementary Table 7, Supplementary Data 1).

Pollination: Bee-pollinated plants evolved 1.07-fold bigger flowers, with changes in floral scent observed in benzyl nitrile (1.23-fold) and in phenylethyl alcohol (0.79-fold), as well as lower amounts of leaf glucosinolates (0.92-fold), such as gluconapin (0.91-fold), glucoraphanin (0.78-fold), glucobrassicin (0.78-fold), and gluconasturtiin (0.64-fold; Fig. 2, Supplementary Tables 6, 7, Supplementary Data 1).

Herbivory: Plants experiencing aphid-herbivory evolved smaller petal width (0.96-fold) and sepal length (0.97-fold). Plants that experienced aphid-herbivory also evolved distinctive floral scent emission with variations in indole (1.20-fold), 1-butene-4-isothiocyanate (0.79-fold), and phenylethyl alcohol (0.78-fold; Supplementary Tables 6, 7, Supplementary Data 1).

Pollination × soil: Plants evolving in limestone soil with bee-pollination evolved smaller plant size (0.91-fold), lower flower production rate (0.90-fold) with lower nectar amount (0.82-fold), and smaller leaves (0.89-fold) but with higher amounts of leaf methoxyglucobrassicin (1.17-fold). In contrast, plants that evolved in tuff soil with bee-pollination evolved to stand taller (1.05-fold), with higher flower production rate (1.05-fold), and bigger leaves (1.16-fold) but with lower leaf methoxyglucobrassicin (0.57-fold; Fig. 2, Supplementary Tables 6–8, Supplementary Data 1). In addition, plants in limestone soil with bee-pollination evolved lower 2-aminobenzaldehyde emission (0.88-fold), but higher leaf neoglucobrassicin production (1.66-fold) than plants in tuff soil with bee-pollination (Table 1, Supplementary Table 7, Supplementary Data 1). In plants that evolved in tuff soil with bee-pollination, we observed a later flowering time (1.10-fold) compared to plants in limestone soil with bee-pollination, as well as an increased emission in methyl salicylate (1.25-fold), and in (*E,E*)-α-farnesene (1.24-fold), whereas leaf glucosinolates were characterized by lower amounts in hydroxyglucobrassicin (0.81-fold) and methoxyglucobrassicin (0.49-fold; Supplementary Tables 7, 8, Supplementary Data 1).

Herbivory × soil: Among the two soil evolutionary lines, aphid-herbivory had a very different effect on plant evolution (Supplementary Table 8, Supplementary Data 1). In plants evolving in limestone soil, herbivory mainly affected the plants' floral scent emission with variations in methyl anthranilate (1.89-fold), 2-aminobenzaldehyde (1.31-fold), (*E,E*)-α-farnesene (0.72-fold), phenylethyl alcohol (0.69-fold), benzaldehyde (0.64-fold), and 1-butene-4-isothiocyanate (0.63-fold; Supplementary Table 8, Supplementary Data 1). In addition, we observed a 1.14-fold change in plant branching and 0.35-fold lower leaf glucoalyssin amount (Supplementary Table 8, Supplementary Data 1). When plants evolved in tuff soil, aphid-herbivory affected mostly the evolution of plant morphology with 0.88-fold smaller leaf, reduced plant height (0.91-fold), and reduced flower production rate (0.87-fold), but increases in *p*-anisaldehyde (1.52-fold), and benzaldehyde emissions (1.38-fold, Supplementary Table 8, Supplementary Data 1).

Pollination × herbivory × soil: Plants that evolved in limestone soil with bee-pollination and aphid-herbivory increased their production in 4 aromatic compounds (2.96, 1.98, 1.43, 1.13 for methyl anthranilate, methyl benzoate, methyl salicylate, phenylacetaldehyde, respectively) and produced 1.38-fold more leaf neoglucobrassicin than plants that evolved in absence of herbivory but with bee-pollination (Supplementary Tables 6, 8, 9, Supplementary Data 1). In contrast, plants that evolved in tuff soil with aphid-herbivory and bee-pollination were smaller (0.88-fold), with flowers emitting 0.34-fold lower (*Z*)−3-hexen-1-ol-acetate than bee-pollinated plants that evolved without herbivory (Supplementary Tables 6–9, Supplementary Data 1).

**Table 1 | Effects of bee-pollination, aphid-herbivory, soil type and their interactions on the evolution of plant morphology, nectar, floral scent and leaf glucosinolates**

| Trait | N | Pollination (P) | Soil (S) | Herbivory (H) | (S) × (H) | (S) × (P) | (P) × (H) | (S) × (P) × (H) |
|---|---|---|---|---|---|---|---|---|
| *Morphology* | | | | | | | | |
| Height day 20 (cm) | 575 | Y (–) | Y (T–) | | | Y | | Y |
| Height day 30 (cm) | 575 | | Y (T+) | Y (–) | Y | Y | Y | |
| Time to flower (day) | 575 | Y (+) | Y (T+) | | | Y | | |
| Leaf size (cm2) | 575 | Y (+) | Y (T+) | Y (–) | | Y | | |
| Flower production rate (per day) | 575 | | Y (T+) | | Y | Y | | |
| Flower number | 575 | | Y (T+) | | Y | Y | | Y |
| Nectar amount (nL/flower) | 564 | | Y (T+) | | | Y | | |
| Flower diameter (cm) | 564 | Y (+) | | | | | | |
| Petal length (cm) | 564 | Y (+) | | | | Y | | |
| Petal width (cm) | 564 | Y (+) | | Y (–) | | | | |
| Sepal length (cm) | 564 | | Y (–) | | Y | | | |
| Stamen length (cm) | 564 | | Y (T+) | | | | Y | |
| *Bioassays* | | | | | | | | |
| Number of bee visits | 575 | | Y (T+) | Y (–) | Y | | Y | |
| *Floral scent (pg.l$^{-1}$.hr$^{-1}$.flower$^{-1}$)* | | | | | | | | |
| Benzaldehyde | 564 | | Y (T+) | | Y | | | Y |
| Methyl benzoate | 564 | Y (+) | Y (T–) | | | | | Y |
| Phenylethyl alcohol | 564 | Y (–) | Y (T–) | Y (–) | | | | |
| Phenylacetaldehyde | 564 | Y (–) | Y (T–) | Y (–) | | | | Y |
| 2-Aminobenzaldehyde | 564 | | Y (T–) | | Y | Y | | |
| (E,E)-α-Farnesene | 564 | | Y (T+) | | Y | Y | Y | |
| (Z)-3-Hexen-1-ol-acetate | 564 | | | Y (+) | | Y | Y | Y |
| Methyl anthranilate | 564 | | Y (T–) | Y (+) | Y | Y | | Y |
| Benzyl nitrile | 564 | Y (+) | Y (T–) | | | | | |
| Indole | 564 | | | Y (+) | | | | |
| Total N-containing volatile compounds | 564 | | Y (T–) | | Y | | | Y |
| 1-Butene-4-isothiocyanate | 564 | | Y (–) | | | | | |
| Total emission | 564 | | Y (T–) | | Y | | | Y |
| *Leaf glucosinolates (pg.mg$^{-1}$)* | | | | | | | | |
| Glucoraphanin | 485 | Y (–) | | | | | | Y |
| Glucoalyssin | 500 | | Y (–) | | Y | | | |
| Gluconapin | 512 | Y (–) | Y (T+) | | | | | |
| Glucobrassicanapin | 509 | Y (–) | Y (T+) | | Y | | | |
| Glucobrassicin | 510 | Y (–) | | | | | | |
| Neoglucobrassicin | 511 | | Y (T–) | | | Y | | Y |
| Gluconasturtiin | 512 | Y (–) | Y (T+) | | Y | | Y | |
| Methoxyglucobrassicin | 511 | | Y (T–) | | | Y | | |
| Total glucosinolates | 477 | Y (–) | Y (T+) | | | | | |

Only traits with a significant effect (indicated by Y for "yes") of either bee-pollination, aphid-herbivory, or soil are shown in this table (see Supplementary Table 7 for full statistical values); (+) indicates a positive effect (increase) of the factor on trait evolution whereas (–) indicates a negative effect (decrease) on plant trait evolution. For soil type, (T–) indicates that tuff lines evolved lower trait values than limestone lines, whereas (T+) indicates higher trait values for plants that evolved in tuff. Only plants of generation 10 were included in this analysis.

## Overall patterns of divergence between soil types

To study the phenotypic differences between soil types, we analyzed F-values of univariate ANOVAs, and multivariate linear discriminant function analyses (DFA). The F-values were calculated for 33 traits of plants that evolved in different soil types, with larger F-values indicating stronger differentiation. The analyses showed that the factor "pollination" led to higher F-values between soil types in plant morphological traits only (Fig. 3; Table 2, Supplementary Data 2). In support of this, using multivariate DFA with 12 principal components summarizing all plant traits, we showed that the divergence between soil types was stronger when plants had evolved with bee-pollination as compared to hand-pollination (Fig. 4, Supplementary Data 3).

## Evolution of plant attractiveness

Because we documented strong evolutionary changes in plant traits driven by soil, pollinators, and herbivory, we investigated how these changes affected plant attractiveness to pollinators and herbivores, by performing choice tests with plants of each treatment group and generation one (i.e. 9 plants were assayed together). In terms of the plants' attractiveness to bees, we found that plants that evolved in tuff soil with bee-pollination without herbivory were more attractive than plants of all other treatment groups (Fig. 2, Supplementary Data 1). There was no statistical difference in attractiveness between the other plant groups. We did not find significant variation in plant attractiveness to alate aphids among our treatment groups.

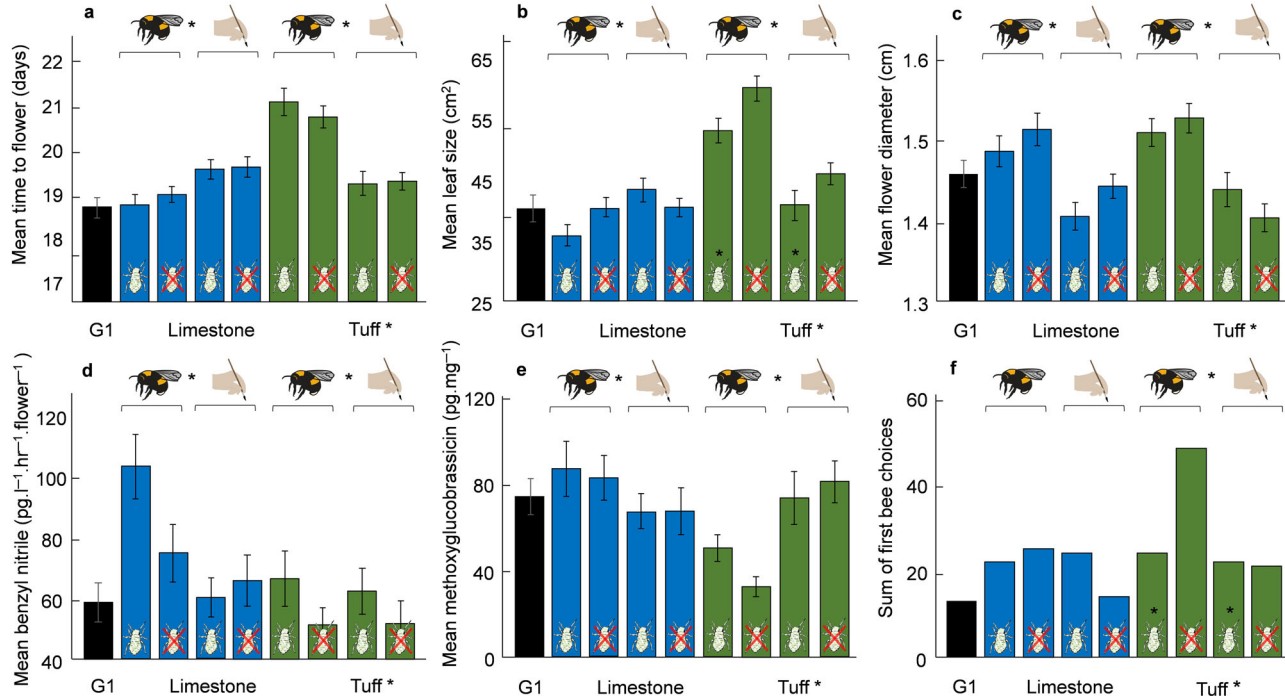

**Fig. 2 | Impact of bee-pollination and aphid-herbivory on plant evolution.**
Evolution of phenology and morphology (**a**: time to flower (*N* = 647 plants), **b**: leaf size (*N* = 647 plants), **c**: flower diameter (*N* = 636 plants)), floral scent (**d**: benzyl nitrile emission (*N* = 635)), leaf glucosinolates (**e**: methoxyglucobrassicin (*N* = 576 plants)), and plant attractiveness (**f**: plant attractiveness to bumblebees (*N* = 213 bees for 639 plants)) in plants of generation 10 (i.e. after 8 generations of selection) in different soil types and plants of generation 1 (G1, i.e. before selection). The effects of ecological factors (soil, pollination, herbivory) on evolutionary changes were estimated by two-sided linear mixed models (LMM). Asterisks indicate significant effects of the factors soil, herbivory, or pollination (*P* < 0.05; detailed statistical values are shown in Supplementary Table 6). Bars show mean trait values (±s.e.m.) among the different treatments.

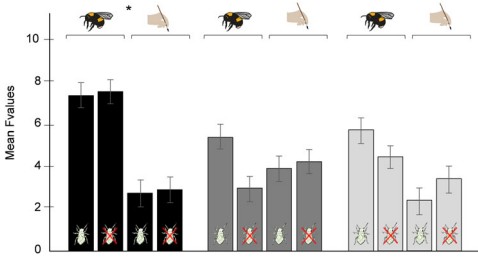

**Fig. 3 | Morphological traits show greater differences among plants evolving in different soil types, with bee-pollination.** The graph shows the effect of pollination (bee-pollination or hand-pollination) and herbivory (aphid-herbivory or no herbivory) on phenotypic differences across different trait categories (plant morphology (black bars), floral scent (dark grey bars), and leaf glucosinolates (light grey bars)), between plants that evolved in different soil types. Differences were measured by F-values of univariate ANOVAs (*N* = 264): F-values were calculated between treatment groups with the same biotic interactions, growing on different soil types. Thus, greater F-values indicate more differentiation between traits of plants that evolved in different soil types (see Table 2 for statistical details). Bars show mean F-values (±s.e.m.) for the three trait categories.

## Evolutionary rates
Plants that evolved with bee-pollination and in tuff soil showed the highest evolutionary rates, with haldanes (=s.d. per generation) being on average 1.36-fold higher than in plants evolving with hand-pollination or in limestone soil (Fig. 5, Supplementary Tables 2, 3, Supplementary Data 4).

## Discussion
The identification of mechanisms driving among-population divergence in plant traits remains one of the most important research questions in plant ecological speciation[9,32]. It has long been known that soil can affect plant evolution through imposing selection on mechanisms for nutrient gain, potentially leading to edaphic specialization and the formation of plant races[1,8,78]. We show here that soil-type also impacts the way pollinators shape plant evolution, and that the interaction between bee-pollination and soil-type causes the strongest phenotypic divergence after 8 generations of experimental evolution. Thus, our data show that soil and bee-pollination can drive rapid plant divergent evolution without a shift in pollinators, thereby affecting not only floral phenotype, but a range of traits related to reproduction, growth, as well as defense. This "soil-pollinator effect" may be caused by i) trade-offs between growth, defense and pollinator attraction; such trade-offs should be more pronounced in nutrient-limited environments, thus slowing down pollinator-driven adaptation. Indeed, in our experiment, plants in the more nutrient-limited limestone soil showed less adaptive evolution and divergence in response to bee-pollination. In addition, ii) different patterns of phenotypic plasticity caused by different soils can change patterns of pollinator-mediated selection, thus also impacting trajectories of adaptive evolution[33]. We argue that the soil-pollinator effect deserves more attention as a potential driver of phenotypic divergence in plants, and perhaps as a starting point and/or enhancer for subsequent shifts in pollinators leading to the evolution of pollination ecotypes.

Whereas the role of individual ecological factors for plant divergent evolution has been investigated in considerable detail, interactions among factors have generally received less attention[61,79,80]. In our study, both the effects of herbivory and pollination on plant evolution were soil-dependent, highlighting the importance of considering multiple factors for trait evolution in plants[79,81]. The importance of herbivory × soil interactions for the evolution of plant defense has been conceptualized in the "growth-defense trade-off", which has received some attention in previous research. Whereas this hypothesis

predicts that plants in low-nutrient soils evolve higher defenses[48,55,82], we did not find higher amounts of glucosinolate defense compounds in plants that evolved in limestone soil (the poorer soil type, see ref. 33). Plants that evolved in limestone soil with aphid-herbivory and bee-pollination actually showed the lowest production of glucosinolates. A reason for this may be that the specialist herbivore used in our experiment does not efficiently select for increased glucosinolates,

## Table 2 | Effects of bee-pollination and aphid-herbivory, and their interaction on divergence of traits in plants that evolved in different soil types

| Trait | N | Factor | Df | $\chi^2$ | P |
|---|---|---|---|---|---|
| *F* values | 264 | **Pollination (P)** | **1** | 8.42 | **0.004** |
| | | Herbivory (H) | 1 | 0.24 | 0.624 |
| | | Trait groups (Tg) | 2 | 1.93 | 0.380 |
| | | Replicate | 1 | 1.36 | 0.244 |
| | | (P) × (H) | 1 | 1.19 | 0.276 |
| | | **(P) × (Tg)** | **2** | 6.33 | **0.042** |
| | | (H) × (Tg) | 2 | 0.25 | 0.882 |
| | | (P) × (H) × (Tg) | 2 | 0.73 | 0.694 |
| Post hoc tests | | | | | |
| Parameters | | Treatment groups | N | *t*-values | P |
| **Morphology** | | **Bee-pollination versus Hand-pollination** | **80** | **3.53** | **0.006** |
| Floral scent | | Bee-pollination versus Hand-pollination | 104 | 0.06 | 0.999 |
| Leaf glucosinolates | | Bee-pollination versus Hand-pollination | 80 | 1.50 | 0.663 |

The table shows the results of a two-sided general linear mixed model with *F*-values as dependent variable (*N* = 264), ecological variables, trait groups (morphology, scent, glucosinolates, see Fig. 3), and their interactions as fixed factors, and replicate as random factor. *F*-values were calculated for plants that evolved in different soil types by one-way ANOVAs for each trait. *F*-values in this analysis correspond to degree of differentiation in traits. Bold indicates significant factors (P < 0.05).

because of its tolerance against aliphatic and to some degree also indolic glucosinolates[83]. In addition, animal pollinators are thought to select against chemical plant defenses, because the presence of defensive compounds in nectar or pollen may make floral rewards less palatable to them[81,84,85]. Indeed, in our study, bee-pollination led to a reduction in the amounts of most glucosinolates, in both soil types. This shows, as suggested by earlier studies[61,85,86] that biotic pollination can lead to lower degrees of chemical defense, when selection for elevated defense chemistry is absent or weak.

Although aphid-herbivory in our experiment did not lead to consistently higher glucosinolate production, it still compromised the evolution of flower attractiveness to pollinators. This confirms earlier experimental evolution studies using chewing herbivores[61], as well as numerous single-generation studies that showed decreased flower attractiveness as a pleiotropic effect of herbivore-induced plasticity[59,80,87]. In our study, aphid-herbivory likely decreased flower attractiveness through diminishing floral scent emission and plant height, that both signal plant attractiveness to pollinators in the here used as well as in other plant-pollinator systems[31,88].

Different patterns and strengths of trait evolution among plant populations lead to population-level differences in plant phenotypes, called plant races, or ecotypes, in case population differences are also associated to local adaptation[89]. In our experiment, the combination of soil difference and bee-pollination led to the strongest divergence among our experimental populations. Whereas both soil and pollinators are considered key drivers of plant diversification[62,90,91], the interaction between the two factors has received little attention, despite closely related plants growing on different soil types often differ in floral traits[62,92–95], suggesting a link between soil- and flower divergence. Van der Niet[63] interpreted the combined divergence between soil specialization and floral trait divergence as the outcome of reinforcement, because this pattern was more pronounced when plants occurred in sympatry than in allopatry. In our experiment, the mechanism for soil × pollinator interactions driving divergence are likely twofold. First, similar to the growth-defense trade-off, plants under pollinator-mediated selection face different trade-offs when

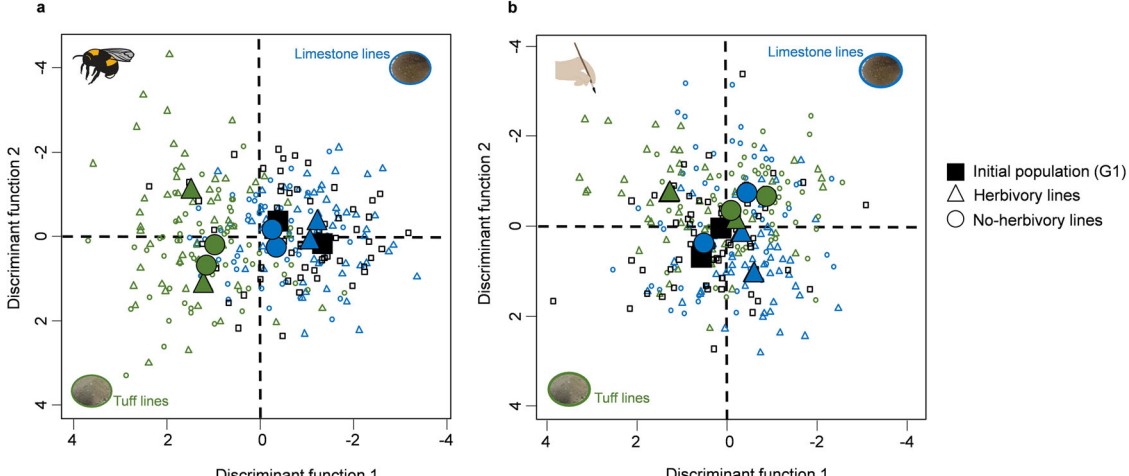

**Fig. 4 | Bee-pollination leads to stronger divergence of plants growing different soil types.** The figure shows multivariate comparisons of plants before and after experimental evolution (plants of generation 1 (initial population) and 10), with **a** plants that evolved with bee-pollination (*N* = 221) and **b** plants that evolved with hand-pollination (*N* = 219), using linear discriminant function analyses (DFA). Blue dots: plants that evolved in limestone, green dots: plants that evolved in tuff; circles: plants that evolved without herbivory, triangles: plants that evolved with aphid-herbivory, black squares: plants of generation 1; filled, enlarged symbols are group centroids of replicates. In bee-pollinated treatment groups, all comparisons between soil types were significantly different, whereas in the hand-pollinated

treatment groups, only plants that evolved in tuff without herbivores were significantly different from limestone lines (see post hoc comparisons between evolutionary lines in Supplementary Table 4). We used replicates of all five treatment groups as factor in the DFA leading to 10 groups for the hand-pollination and bee-pollination group, and 9 discriminant functions. (*N* = 283 for bee-pollinated plants and generation one: functions: 1–9 $\chi^2$ = 445.00, 2–9: 241.46, 3–9: 166.53, 4–9: 103.63, all *P* < 0.001, 5–9: 50.95, 6–9: 29.81, 7–9: 13.39, 8–9: 5.14, 9: 0.58 with *P* > 0.05, *N* = 281 for hand-pollinated plants and generation one: functions: 1–9 $\chi^2$ = 327.12, 2–9: 237.02, 3–9: 148.10, 4–9: 42.90 all *P* < 0.001, 5–9: 48.23, 6–9: 30.22, 7–9: 16.39, 8–9: 8.98, 9: 2.50 *P* > 0.05).

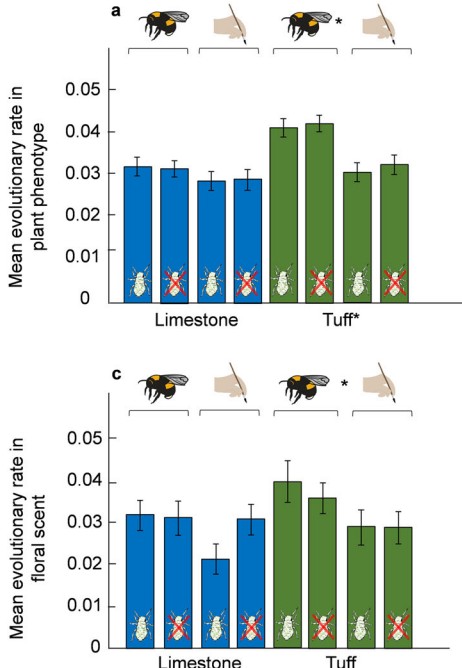

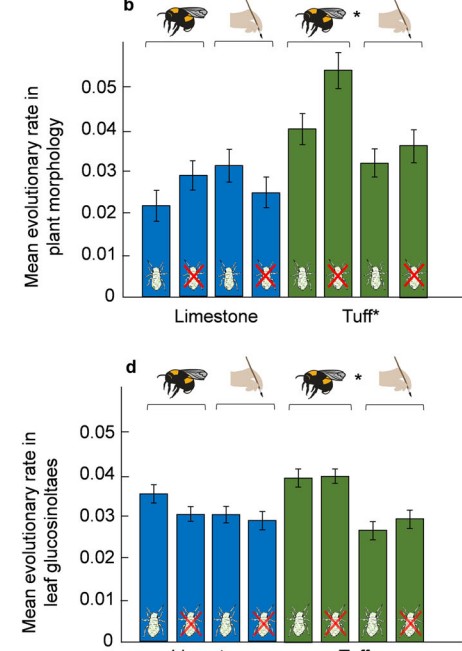

**Fig. 5 | Tuff soil and bee pollination leads to higher evolutionary rates.** The graphs show the impact of pollination (bee-pollination or hand-pollination), herbivory (aphid-herbivory or no herbivory), and soil (tuff or limestone) on evolutionary changes between generation one and ten (i.e. evolutionary rate), measured in absolute values of haldanes (=s.d. per generation). **a** overall phenotypic change, **b** plant morphology, **c** floral scent and **d** leaf glucosinolates. Blue bars: plants that evolved in limestone soil, green bars: plants that evolved in tuff soil. Note that bars represent absolute evolutionary changes (mean ±s.e.m.) and indicate evolutionary rates of different traits, so similar bar height does not necessarily indicate evolutionary changes in the same direction or similar evolutionary rates in the same traits. Asterisks indicate significant difference between the factors pollination, herbivory and soil ($P < 0.05$, see Supplementary Tables 2 and 3 for statistical details).

growing on soils with different nutrient availability. In low-nutrient soil, such as our limestone soil, a decrease in above-ground traits, as observed in our study, yet not predicted by pollinator-mediated selection, may result from biomass trade-offs, where plants allocate more resources into below-ground and less into above-ground biomass[96]. Such reallocations allow for an optimal nutrient investment, as they lead to a direct fitness benefit or an increase in overall resource uptake[97–99]. When nutrients are less limited, as in our tuff soil, such trade-offs may be less important and plant evolutionary trajectories may follow more directly selection imposed by pollinators. Secondly, and in addition to trade-offs, Dorey and Schiestl[33] showed that pollinator-mediated selection can change depending on patterns of phenotypic plasticity induced by different soil types. The patterns of divergence documented here are likely the outcome of the combined effects of different selection and different trade-offs as a consequence of growing on different soil types.

Floral diversification has been assumed to be tightly connected to either quantitative or qualitative shifts in pollinators, typically driven by pollen limitation[10,11,31,100–103]. Nevertheless, flowers may also diversify without pollinator shifts[104], and mechanisms underlying this phenomenon may relate to different use of the same pollinators, abiotic factors, antagonists, or plant community composition[68,81]. Our study adds the soil-pollinator effect to the mechanisms leading to floral (and general plant) divergence without pollinator shifts. As the soil-pollinator effect forms a starting point for population divergence, it may subsequently be followed by a shift in pollinator guilds, as traits related to pollinator attraction (such as floral scent and petal size) or pollinator-flower fit (such as flower- or reproductive organ size) diverge between populations. The consequence may be a change in pollinator effectiveness (sensu[78]), thus further reinforcing divergent selection caused by different pollinators attracted, and/or their different efficiency in flowers of plants growing on different soil types.

In this way, the soil-pollinator effect may form the starting point for pollinator-driven plant diversification, however, independent of a geographic pollinator mosaic.

## Methods
### Study system
In this experiment we used annual "fast cycling" *Brassica rapa* plants (Fast Plants® Standard Seed with maximum genetic diversity). These plants are fully outcrossing (self-incompatible), have short generation time, harbor a high standing genetic variation and have previously been shown to be suitable for selection- and experimental evolution studies[31,61,76,105].

### Experimental design
In 2018, 440 seeds of fast cycling *Brassica rapa* obtained from Carolina Biological Supply (Burlington, NC, USA) were sown out on standardized soil (commercially obtained soil with a homogeneous texture and distribution of nutrients) in a phytotron under standardized humidity, temperature and water conditions. Of these 440 seeds, 410 germinated and were used to produce full sib seed families by artificial crossings. Artificial crossings consisted of manually crossing 205 randomly assigned plant pairs, which resulted in a total of 163 seed families (only pairs where both parents produced seeds were used as seed families). Of these 163 families, 98 families were randomly selected for use as the starting population of our experimental evolution.

### Experimental evolution
Our experimental evolution included 8 different treatments, which encompassed a full factorial combination of three different factors: growing in either limestone- or tuff soil (see Supplementary Table 11 for chemical and physical properties of each soil type), hand-pollination or bumblebee-pollination, with- or without aphid-herbivory (Fig. 1). The 98 families were randomly assigned to two

replicates (A and B) and 8 treatment groups, so that each replicate included 49 plants (allowing for a squared 7 × 7 setup during pollination). Plants were first grown in a phytotron under 24 h of light, 21 °C, 60% humidity and were watered once a day (at 8:00 h). After repotting, we moved the plants to an air-conditioned greenhouse with natural and additional artificial illumination to achieve growing conditions with 16 h of light and a constant temperature of 23 °C, with uniform watering. The evolution experiment lasted for 8 generations with insects, followed by two more generations without any insects, including one in a common garden where plants were grown in standardized soil. This was done to avoid plasticity due to herbivory and to reduce maternal effects carried throughout the experiment (e.g., in aphid-herbivory groups). Because no sign of inbreeding was found after 8 generations (see below), replicates were not crossed but kept as isolated lines during the 10 generations to be able to assess independent and reproducible evolutionary changes. Inbreeding effects were assessed after eight generations of selection by crossing plants within or between replicates. We showed that plants crossed between replicates produced a similar amount of seeds than those gained from within-replicate crosses (ANOVA: $F_{271} = 0.05$, $P = 0.817$, Supplementary Data 5). Plants grown from the seeds of the crosses within and between replicates also showed similar phenotypes (MANOVA including five traits that were analyzed: $F_{254} = 1.58$, $P = 0.167$, Supplementary Data 5), suggesting little or no inbreeding effects on plant phenotypes.

## Selection treatments

Our evolution experiment was set up to mirror natural conditions found in natural Southern Italian populations of *Brassica* spp., especially *B. incana*, a close relative to *Brassica oleracea and B. rapa* (Arrigo and Schiestl, unpublished data). The Mediterranean region, and especially Southern Italy is a center of diversity for *Brassica* spp. with three species strictly endemic to this region[106]. Both soil types used throughout the experiment were collected in *Brassica incana* populations in Campania (Italy) and represent two of the major types (tuff, limestone) of soil on which *Brassica* spp. grow (Arrigo and Schiestl, unpublished data). Soil was collected in July and October 2018, at Valico di Chiunze (40.719°N, 14.619°E) for limestone, and at Monte di Procida (40.809°N, 14.045°E) for tuff soil. We collected soil from the surface layer (0–15 cm, ca. 500 kg per soil type), sifted it (using a mesh of 1 cm of diameter), and stored it in textile bags to be shipped to Switzerland. The chemical composition of the soils was analyzed (see ref. 33). The soil was not sterilized before use. Because at generation three and four our experimental *B. rapa* plants produced few seeds, from generation five onwards we added fertilizer to both soil types in the form of 10 mL of an universal garden fertilizer diluted in 10 L of water (NPK: 8-6-6 with traces of B, Cu, Fe, Mn, Mo, Zn; Wuxal, Maag Agro, Dielsdorf, Switzerland). These nutrient additions were first performed during sowing and then a second time after pricking. Because resource limitation is likely an important component of soil-driven evolution, we analyzed resource limitation after adding fertilizer. We showed that reproduction and growth in our experimental plants was still resource limited even with fertilizer, as plants produced fewer flowers, fewer seeds, smaller flowers as well as had a lower growth rate than plants grown on standardized soil with optimal fertilization (Supplementary Fig. 1, Supplementary Data 6).

## Herbivory

Plants were exposed either to pre-flowering aphid-herbivory (*Brevicoryne brassicae*) or kept uninfested (as indicated by the symbols, H: herbivory, NH: no herbivory). *Brevicoryne brassicae* is a notorious agricultural pest that specializes on Brassicaceae by sequestrating glucosinolates for self-defense. In the wild, *Brevicoryne brassicae* is one the most common herbivores on crucifers such as *Brassica incana* (Arrigo and Schiestl, unpublished data). To start an aphid colony, we collected individuals from different plants at the Botanical Garden of

the University of Zurich (Switzerland) in the summer of 2018 and reared them in a climatic chamber on unselected lines of fast cycling *Brassica rapa* plants under 16 h lights at a temperature of 23 °C and humidity of 70%. In our evolution experiment, aphid-herbivory started at "the two true leaves stages", which occurred around 13–14 days after sowing out for plants growing in tuff and 17–18 days after sowing out for plants growing in limestone. Each individual plant was infested on the leaves with 10 wingless individuals, which were allowed to feed and reproduce on plants for 72 h. Previous studies showed that 10 aphids are enough to induce strong plastic responses in plants within the first hours of the infestation[107] and to lead to different levels of infestation depending on plant defenses[108]. To avoid aphid migration to more suitable plants, infested-plants as well as non-infested plants were covered by a net (7 × 7 × 15 cm) with mesh size of 680 μm (Bugdorm, model DC0901-W). After the three days of herbivory, nets and aphids were carefully removed from the plants and from this day on were checked every day for left-over aphids, which were subsequently removed as well.

## Pollination

For pollination, we either hand-pollinated plants or used bumblebees (*Bombus terrestris*[61]) for pollination (as indicated by the symbols, H: hand-pollination, B: bee-pollination). *Bombus terrestris* is a common and efficient pollinator of crucifers and is also one of the most common pollinator of populations of *Brassica incana* (Arrigo and Schiestl unpublished data[109,110];). For all pollination including pollinator bioassays at the end of the experiment, we used only bumblebee workers. Bees were purchased in Switzerland (Andermatt Biocontrol Suisse AG) and hives were kept inside the greenhouse in a flight cage (l × w × h: 75 × 75 × 115 cm). One to two colonies of *B. terrestris* were used every generation for pollination, and new colonies were used for each generation. Because bumblebees had not experienced any plants or flowers before being delivered, we fed them for a week with *Brassica rapa* "fast cycling" plants to gain experience with these flowers. In addition to plant flowers, bumblebees were fed with supplemental pollen (Biorex, Ebnat-Kappel, Switzerland) and sugar water (Biogluc sugar solution, Biobest). To avoid bias in pollinator choice, "feeding plants" were from unselected lines and were grown on standardized soil. Three days before pollination and choices test, we removed all plants from the cages and supplemental pollen and nectar solution were the only source of food left for bumblebees. To enhance bumblebee's foraging activities, bees were starved by removing all food 16 h before pollination and choices tests. Pollination as well as bee choice tests were done 10 days after herbivory (when all aphids had been removed) and when most of the plants were flowering simultaneously on both soil types. Pollination was performed between 8.30am and 5.30 pm for each replicate and in each generation. For pollination, plants were randomly set up in a square of 7 × 7 in a flight cage (l × w × h: 2.5 m × 1.8 m × 1.2 m) with a distance of 20 cm between plants. A total of seven bee workers were released individually and sequentially, and recaptured after visiting a total of five different plants; each bee was only used once throughout the experiment. As a consequence of bee-pollination, for each replicate in each generation, around 17 plants produced seeds (replicate A, mean: 17.41 ± 2.63; range of 11–23, replicate B, mean 16.57 ± 2.27; range of 12–22). The number of bee visits was limited to ensure pollen limitation and thus pollinator-mediated selection. The level of pollen limitation at the replicate level in our experiment was considered at the upper range of pollen limitation in natural environments, where many more plants interact with fewer pollinators[111,112]. For the hand-pollinated plant groups, we randomly selected 28 plants per replicate for pollination to achieve a similar proportion of pollinated plants. In these control groups, we randomly assigned one father plant to a mother plant of the same replicate, sampled one long stamen that we used to deposit an excess of pollen on stigmas of four flowers per mother plant. Because in the

bee-pollinated treatments, visited plants had an average visitation (bees per plant) between 1 and 2 (mean ± s.d.: TNHB: 1.55 ± 0.13; THB: 1.78 ± 0.22; LNHB: 1.66 ± 0.23; LHB: 1.71 ± 0.25), a single father plant for the hand-pollinated treatments was considered leading to similar average number of paternities compared to the bee-pollinated treatments. Because in *Brassica rapa* flowering occurs in acropetal succession, we marked the lowest and highest open flowers of inflorescences of visited plants and only fruits that developed from between these marks were collected and counted after ripening. After pollination, plants were kept in the same standardized conditions for four weeks after which plants were deprived of water, to dry for seed maturation. Once dried, fruits and seeds of the different treatment groups were harvested, counted, and weighted. Seed set per plant was then used to calculate the contribution of each plant to the next generation, to ensure that all individuals contributed proportional to their seed set in the replicate to the next generation. Plant relative seed set, calculated as the seed production of one individual divided by the mean seed set of the replicate, was used for calculating a plant's contribution to the next generation as it indicates the contribution of each individual to the total seed set pool in the replicate. Seed contribution defined as relative seed set was adjusted to achieve a sample size of 49 plants per replicate. The seed contribution of each pollinated plant into next generation can therefore be summarized with the following formula:

$$\text{Plant contribution} = \frac{\text{Individual seed set}}{\text{Replicate sum of seeds}} * 49 \qquad (1)$$

### Evolutionary changes
Because our experimental plants did not experience any phenotypic signs of inbreeding effects, in generation 9, we manually and randomly cross-pollinated all plants within each replicate, so that each plant was a pollen donor and also a pollen recipient. Of the resulting seeds (generation 10), we picked randomly one seed of 36 families to obtain 36 plants per replicate per treatment. We also selected 36 families of the ancestral population (generation one) for each replicate. To assess evolutionary changes, plants of generation 10 were grown simultaneously with generation one (i.e. as a resurrection experiment) on standardized soil (with optimal fertilization; Patzer Erden, Sinntal Germany) and in standardized greenhouse conditions.

### Plant traits
To assess evolutionary changes in plants at the end of the experiment, we measured plant morphology, floral volatile emission (scent), and leaf glucosinolate content in plants of generation one and ten on day 27, three days before bioassays were conducted. All plants were phenotyped on the same day in order to minimize variation between plants due to development. Because volatiles were collected in a non-destructive way (in contrast to some morphology measurements where three flowers were sampled), we performed scent collection first to avoid collecting altered volatile emission from damaged plants. Floral volatiles were collected from plant inflorescences as soon as at least three flowers were open, using headspace sorption with tenax sorbent and a push-pull collection system[33]. Samples were analyzed by gas chromatrography with mass selective detection, identified by comparison of mass spectra and retention times with synthetic reference standards, and quantified using dose-response curves for all compounds. Amounts of scent compounds were calculated on a per-flower basis for statistical analysis. A detailed description of the collection, identification and analyses protocol can be found in[33]. Floral morphology (petal display: petal width, petal length, flower diameter, sepal length, long stamen length, pistil length) was measured by sampling 3 flowers per plant and by carefully placing petals and reproductive organs on a white paper sheet. Sheets were immediately

scanned before flower display started wilting. Mean values of the three sampled flowers were calculated and used in statistical analyses. Floral nectar was collected from the same three flowers using a 1 μL microcapillary (Blaubrand, Wertheim, Germany). Nectar amount per flower was then calculated by dividing the collected nectar volume by the number of sampled flowers. Plant height was measured at two different times of plants growth, once in the early stage when the first plants started to flower (day 20), and a second time on the day when bioassays were conducted (day 30). On the bioassay day, we counted number of flowers, flower production from flowering onset to bioassays day, cumulated secondary branch length, number of flowering branches, as well as leaf traits such as leaf length and -width. From leaf length and leaf width we estimated leaf area according to the following formula: leaf size = (leaf width × leaf length) divided by 2. Flower production rate was calculated using the time to flower and flower production according to the following formula: flower production rate = flower production from flowering onset to bioassays date/flowering duration, where flowering duration represents the number of days a plant had flowered before bioassays.

### Leaf glucosinolates
For leaf glucosinolates analyses, we collected, on day 34, around 50 mg of fresh leaf tissue from 26 to 35 plants of each replicate and treatment of generation ten and one. Leaf tissue was flash frozen in liquid nitrogen to avoid the hydrolysis of glucosinolates by myrosinase. After dry-freezing samples for 48 h (10 bar, −80 °C, alpha 2–4 LSCplus, Christ, Germany), we weighed samples and ca. 20 mg per sample were ground to a fine powder at a speed of 4 m/s with a FastPrep-24™ grinder (MP Biomedicals™) using 3 beaded balls. We added 1 ml of ice-cold MeOH: water (70: 30; methanol HPLC grade) with sinalbin (5 μg ml⁻¹) as internal standard to the freshly ground samples. Samples were vortexed for 5 s and immediately incubated for 10 min at 85 °C in a shaking bloc heater (600 rpm, Eppendorf Thermomixer® comfort; Eppendorf, Hamburg, Germany). To assure full extraction of the leaf material, samples were moved in an ultrasonic water bath for 10 minutes (AL 04-04; Advantage-Lab™, Darmstadt, Germany) and then centrifuged at 20,000 × g for 10 min (Sigma 1–16, Sigma Laborzentrifugen GmbH, Osterode am Harz, Germany). We transferred 0.7 mL of the resulting supernatant to an Eppendorf tube stored at −20 °C for two hours. Because the supernatant was not clear enough, we centrifuged it a second time at 20,000 × g for 10 min and 0.35 mL of the supernatant was transferred into a glass vial which was stored at −20 °C until high-pressure liquid chromatography (HPLC) analysis. We injected 15 μL of supernatant into an Agilent 1260 Infinity II Binary LC system, and compounds were separated on an Agilent Poroshell 120 CS-C18 column (2.7 μm, 2.1 × 100 mm). Compounds were eluted at a constant flow of 0.5 mL/min at 40 °C with an acetonitrile−0.25% formic acid gradient as follows: starting at 2% acetonitrile, 5 min at 30% acetonitrile, 7.5 min at 45% acetonitrile, 8–11.5 min at 100% acetonitrile, and reconditioning for 3.5 min at 2% acetonitrile. To quantify glucosinolates, we recorded ultraviolet (UV) absorbance spectra at 229 nm by a diode array detector, and compared spectra to the peak area of the known concentration of the internal standards. We identified 11 candidate glucosinolate compounds with distinct molecular masses and HPLC retention times. Glucosinolates were quantified using four calibration curves with four levels ($k = 4$) prepared in MeOH: water (70: 30 ratio) at the following concentrations: 1, 5, 10, 20 μg ml⁻¹, for gluconapin (linear curve fit, $R^2 = 0.9994$), glucobrassicanapin (linear curve fit, $R^2 = 0.9989$), gluconasturtiin (linear curve fit, $R^2 = 0.9997$) and sinalbin (linear curve fit, $R^2 = 0.9998$). Calibration curves and quantifications were based on the peak area ratio between each analyte and the internal standard (sinalbin at 5 μg.ml⁻¹). The concentration range was selected to include the expected plant glucosinolate concentration and the calibration curve (gluconapin, glucobrassicanapin, gluconasturtiin, sinalbin) was selected according to glucosinolate side

chain (either indole-glucosinolate, aromatic, thioakyl or alkenyl). We identified ten different glucosinolates in our leaf samples and all were considered for statistical analyses. We identified outliers in some of the glucosinolates compounds and decided to consider these values as missing values rather than excluding whole individuals. Final sample size was therefore varying in a range of $N = 552$ for glucoraphanin to $N = 579$ for indole glucosinolates.

### Pollinator preference assays

To assess differences in plants' attractiveness to pollinators, we performed bumblebee (*Bombus terrestris*) preferences (first choice) assays at the usual pollination date of the tuff-soil evolutionary lines (30 day after sowing out) between 8:30am and 5:00 pm. Each run of the behavioral assays included nine plants, with one individual of each treatment group (generation 10) and one plant of the starting population (generation one, G1) assayed together. The nine plants were set up in a flight cage (l × w × h: 2.5 m × 1.8 m × 1.2 m) under controlled light and temperature conditions, in a squared 3 × 3 set up with a distance of 20 cm between individual plants. To ensure independence between choices and plant position, we changed the plant positions between runs, and we also released the bumblebees at different locations. For each test run, a total of three bumblebees were used to assess plant attractiveness. Bees were released individually and caught immediately after they made their first choice. Each bee was only used once, and a successful choice was recorded only when bees landed on an inflorescence and started to collect nectar and/or pollen. In total 71 runs of bioassays were performed, using a total of 639 plants and 213 bees.

### Aphid preference assays

To assess differences in plant attractiveness to *Brevicoryne brassicae* aphids, we performed aphid preferences assays at the pollination date of the evolution experiment (30 day after sowing out). Only morphs of aphids at the migratory stage (alate = winged) were used in behavioral assays. Every test run included the same nine plants that were previously used for pollinator preference assays (one individual of each treatment groups (generation 10) and one plant of the initial population, generation one). Plant attractiveness to aphids was assessed in a flight cage (l × w × h: 0.9 m × 0.6 m × 0.6 m) under controlled light and temperature conditions where plants were placed in a squared 3 × 3 set up with a distance of 10 cm between plants. For each test run, a total of 10 alates were used to assess plant attractiveness to aphids. Aphids were released by deposing them on the cage in the middle of the square. Because alatae can take 48–72 h after release before making their final choice[113], we only recorded choices after 72 h by recording how many individuals were found on each plant of the test group. Tests were replicated 71 times for a total of 710 aphids and 639 plants used.

### Statistical analysis

**Trait evolution**. Evolutionary divergence in floral traits was evaluated in generation 10 by one-way ANOVA, linear discriminant function analysis (DFA), and linear mixed models. ANOVA was performed to assess the divergence of traits in plants of different soil types in generation 10. ANOVA was done for 33 individual traits in all treatment groups and replicates separately, using "soil" as independent factor. The resulting F values for the 33 traits (Supplementary Data 2) were compared by a general linear model with F-values as dependent variable, herbivory, pollination, and their interaction as fixed factors, and replicate as random factor. Because some traits were highly correlated (Supplementary Fig. 2), we excluded several traits from this analysis. The following traits were excluded because of high (>0.65) correlation coefficients with other variables: height at day 20 ($r^2 = 0.71$ with flowering time), number of flowering branches ($r^2 = 0.76$ with cumulative branches length), petal length ($r^2 = 0.81$ with flower diameter), petal

width ($r^2 = 0.68$ with flower diameter), flowers produced per day ($r^2 = 0.88$ with flower number), phenylethyl alcohol emission ($r^2 = 0.89$ with phenylacetaldehyde), and benzyl nitrile emission ($r^2 = 0.73$ with 2-aminobenzaldehyde). In our selection between correlated terms, we applied a criterion of "significant selection", wherein all traits known to be under significant selection in our model system were kept. We did not use principle components for this analysis as it would have made the interpretation and separation into categories (morphological, floral scent, glucosinolates) difficult.

For a multivariate analysis of trait divergence, we used DFA in each of the pollination treatment groups separately. To perform DFA on uncorrelated variables, we used principle component scores (PCs) in the DFA (Supplementary Data 3). We conducted principal component analysis on all traits ($N = 40$, but excluding total volatile emission and total glucosinolate production) of generation 10 and 1 combined. The principal component analysis resulted in 12 PCs with an eigenvalue above 1 explaining 72.1% of the total variation (Supplementary Table 10).

To study the contribution of treatments (soil, pollination and herbivory) on plant evolution, we used linear mixed model (package lme4), with individual traits as dependent variable, with "replicate", as random factor, and "treatment" and its interaction with replicate as fixed factors (Supplementary Table 5, Supplementary Data 1). Additionally, individual trait values between plants of generation one and ten were also compared using pairwise post hoc comparisons (Supplementary Table 7, Supplementary Data 1).

The impact of "pollination", "herbivory" and "soil", and their interactions on evolutionary changes was evaluated in plants of generation 10 only. We used a linear mixed model (package lme4), with individual traits as dependent variables, replicate as random factor, "pollination", "herbivory", "soil", and their interactions as fixed factors (Table 1, Supplementary Table 6, Supplementary Data 1). To simplify the interpretation of the multiple significant interactions, we also assessed the evolutionary changes within each soil line separately (Supplementary Table 8). The values for all floral scent compounds, leaf glucosinolates, and nectar amount were $\ln(x+1)$ transformed before analysis to approach normal distribution.

We assessed evolutionary rates by calculating the haldane index for synchronic comparisons between evolved lines and ancestral lines[114]:

$$h_{e,a} = \left( \frac{\bar{x}_e}{\sigma_p} - \frac{\bar{x}_a}{\sigma_p} \right) / g, \tag{2}$$

where $\bar{x}_e$ represents the mean trait value for one evolved line (G10), and $\bar{x}_a$ represents the mean value of the same trait for the ancestral population (G1), $\sigma_p$ represents the pooled standard deviation of the evolved and ancestral lines, and g is the number of generations during the experiment (g = 10 for the current study). Evolutionary changes in haldanes for individual traits are reported in Supplementary Table 1 and Supplementary Data 4. Because a single evolutionary line had both negative and positive evolutionary rates, we used absolute values for statistical analyses. Therefore, absolute values of haldanes were used in linear mixed models to assess how "pollination", "herbivory", "soil" and their interactions affected evolutionary rates of different trait groups (morphology, floral scent, leaf glucosinolates). To do so, absolute values of haldanes of each trait was used as dependent variable, "replicate" as random factor, "pollination", "herbivory", "soil", "trait groups" and their interactions as fixed factors (Supplementary Table 2). Because soil was a strong factor driving plant evolution, we later ran linear mixed models within each soil evolutionary line separately. In these analyses, we used absolute values of haldanes as dependent variable, "replicate" as random factor, and "pollination", "herbivory", "trait groups" and their interactions as fixed factors (Supplementary Table 3). These analyses were performed using the

same 33 traits as for the F-value analysis, to reduce the effect of correlations in the overall assessment of evolutionary rates among trait categories.

**Plant attractiveness.** Attractiveness of plants to bumblebees and winged aphids was analyzed using generalized linear model with Poisson distribution, with "first choices" as dependent variable, replicate as random factor and "pollination", "herbivory", "soil", and their interactions as fixed factors (Supplementary Data 1). All statistical analyses were performed with R software 4.0.0 (2020, R Foundation for Statistical Computing, Vienna, Austria). Generally, we did not adjust P-values for multiple comparisons (Bonferroni correction), because we compared individual traits between treatment groups, not groups of traits. For individual traits, the likelihood of type I errors is independent of the number of comparisons done.

### Reporting summary
Further information on research design is available in the Nature Portfolio Reporting Summary linked to this article.

## Data availability
The data generated in this study have been deposited in the ZORA database of the University of Zürich at https://doi.org/10.5167/uzh-257663.

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

## Acknowledgements

We thank Laura Dällenbach, Franz Huber, Rayko Jonas, Markus Meierhofer, Luca Arrigo for their invaluable help with this study. Timotheus Van Der Niet provided helpful comments to an earlier version of the manuscript. The research was funded by the Swiss National Science Funds (SNF grant no. 31003A_172988 to F.P.S.).

## Author contributions

Conceptualization: F.P.S., T.D. Experimental: T.D. Analysis: T.D., F.P.S. Writing: F.P.S., T.D.

## Competing interests

The authors declare no competing interests.
