## [Peer Review File · Nature Communications]

Bee-pollination promotes rapid divergent evolution in plants growing in different soilsReviewers' Comments:

Reviewer #1:

Remarks to the Author:

This manuscript builds upon a series of recent, high impact experimental evolution studies by Schiestl and associates, using rapid cycling *Brassica rapa* plants as a model system to track the factorial effects of pollinators, herbivores, and their various combinations on trait evolution over a surprisingly short period of selection (8-10 generations). These previous studies were important not only for their surprising results, but also for the authors' unbiased analysis of trait combinations, openness to unexpected outcomes and synergistic effects, and the generation of novel ideas for future research that followed logically from their findings.

Thus, I was excited to read the present manuscript, which builds logically on these foundations by adding soil type (both for direct and indirect effects) as an experimental variable. There are good reasons to expect the "soil-pollinator" effect detailed by the authors here, going back to early work by Verne Grant on "pollinator climate" and subsequent critics of Grant's work on the *Aquilegia* system. Edaphic specialization is a major theme in plant diversification, and decoupling soil and pollinators in an experimental design is valuable and overdue.

I would like to separate my remarks into two categories.

1) Effort and impact: the current study summarizes a remarkable amount of work and reveals (I think) an important set of non-random interactions between soil, bee pollination and history of herbivory. Anything more than a cursory glance would reveal the enormous effort invested into a) generational replication of experimental conditions, including hand-pollination and controlled bee- and aphid exposure, b) the large number of standardized measurements, including different kinds of chromatographic separation, and c) highly replicated behavioral assays with bees and aphids. Considering all the work summarized here, much of it undoubtedly performed during COVID-19 lockdowns, it is remarkable that the study has only two authors. The summary figures are so derivative that it is a bit difficult to interpret them at first glance, but if the authors' interpretations are correct, they have identified a potentially fascinating link between biotic and abiotic drivers of plant diversification in a controlled lab experiment. However, I have some concerns about how to interpret soil differences, when the authors added fertilizer after Generation 5 and decided not to autoclave or sterilize the soil before Generation 1. "Soil differences" in this study appear to mean many things... not all of which were controlled. Thus upon closer examination, the study may not be as elegantly designed as one might hope (e.g. beginning with common soil and adding or withholding key macro or micronutrients, microbes, etc.).

2) Outline and presentation: despite my interest in the findings of this study, I found many problems in the current outlay of this manuscript. First, it simply does not work well in the Nature format. I provide numerous examples below of places where the reader does not have sufficient information to understand the motivation or justification for the study, or even how the soil types differed, before the results are presented and interpreted. Some of this is revealed afterwards in the Methods, but by then it is too late. And while I like the idea of replicating some of the natural conditions for *Brassica incana* and its aphid antagonists in southern Italy (rather than continuing to build on the previous studies using *Pieris* larvae as herbivores), this information is lacking in the Introduction, and Figure 5 comes far too late for the reader to appreciate the experimental design. Second, I was disappointed that the authors did not develop or test a priori hypotheses for nutrient tradeoffs, which were only mentioned later in the Discussion. There is a large body of theory that could have guided such hypotheses, especially given the life history parameters of rapid cycling *B. rapa*. Thus, the Introduction comes across as unnecessarily cursory and exploratory. Third, some of the response variables are summarized in highly derived ways, and I wonder about some basic aspects, such as a) how did the authors deal with collinearity of chemical response variables and correlated morphological traits, and b) how did they handle variance measures for individual variables when combining them to create aggregate variables (e.g. for all volatiles or all morphological traits)? I do not understand how the bee and aphid behavioral assays described in the methods translated into results analyzed statistically or summarized in figures/tables. The figures don't show this very clearly, in my opinion. Finally, the manuscript is full of small, distracting editorial mistakes that should have been caught and corrected

before submission. It is a level of sloppiness unsuitable to this (or any) journal.

Below, I provide specific comments and questions.

Introduction: could have used another round of proof-reading, given the number of editorial errors:

Line 36: neutral, not neural.

Line 57: pollinators, in contrast, act...

Line 72: although its effects on floral evolution are less well known, water availability...

Lines 74-75: awkward wording, please reword this passage.

Lines 80-81, Figure 5: This passage is key to setting up your experimental design and expectations for readers but the backwards format of Nature makes it feel like an afterthought here. Figure 5 really should be presented as Figure 1, with more explanation and justification for the reader. Similarly, the passage from lines 390-400 creates a very different and more specific picture for the reader: the authors are intending to replicate natural conditions (e.g. for *Brassica incana*) in Italy, in which limestone vs. tuff soils, bee pollination and aphid herbivory are key ecological variables. At this point in the introduction, the reader has no idea of this. For example, having read previous papers by this group, one might expect the use of *Pieris* butterfly larvae, rather than aphids, as the herbivore treatment. How do limestone and tuff soils differ? What should we expect? Later, the authors mention having performed a component analysis of the soils, but those results are not presented.

Line 81: soil, pollination and herbivory

Line 87: and if so, which combinations lead to...

Lines 88-90: "Initially, we predicted that more factors should cause more divergence, and thus the interaction of both herbivory and pollination lead to the strongest divergence among plant populations on different soil types"

This is a weak closing sentence to a generally strong introduction. Did the authors not have more specific hypotheses and predictions to test? I appreciate the exploratory nature of this approach, but I was expecting to see a stronger a priori conceptual or theoretical component framing alternative outcomes here. For example, if soil N is limiting, would that constrain responses to selection for floral display (flower size, number) or scent (e.g. lines 102-103), and would this be exacerbated when under attack by herbivores (thus increasing demand for N-related defensive chemistry)? Again, as above, the reader does not yet know what the soil treatments are. One could imagine starting with a generic potting soil and manipulating micronutrients, particle size or pH, in a controlled fashion.

Results:

Lines 96-97: "plants that evolved in tuff" – I understand what the authors are saying here, but it is confusing to the reader given the possibility that they are comparing plants whose seeds were collected in locations with tuff vs. limestone soil. Instead, they could write "plants that experienced tuff as a soil treatment..."

Line 120: revise with punctuation

Line 143: the factor "pollination"

Line 140-147: how many of these traits were correlated?

Line 160: first mention of "haldanes" as a unit: please define in parentheses.

Line 162: in categories, with morphological....

Figure 1: the icons above and within each bar are too small to be appreciated when viewing this figure. Vertical axes are lacking units in panels C, D, E and F. Please provide them directly or in the figure legend. Please explain what G1 means in the figure legend. Perhaps also mention what was done to accommodate multiple comparisons in this figure.

Figure 2: please explain briefly in the figure legend that these are aggregate traits. How are the error bars reflective (or not?) of variance across the combined traits?

Figure 3 legend: lines 204-205, grammar: please check treatments and groups (singular vs. plural). Please add one sentence to the legend helping to interpret the results summarized herein, rather than simply referring the reader to Table S4. The limestone and tuff-treated lines segregate left to right (DF1) in the bee pollination panel and to a lesser extent in the hand pollination panel, but not top to bottom (DF2) in either panel. How might you concisely summarize this outcome?

Figure 4, panel C especially: one might conclude that "limestone, hand pollination, no herbivory" leads

to the same outcome as “tuff, bee pollination and herbivory” for mean evolutionary rates regarding floral scent, but this is due to selection for different scent compounds (lines 130-136).

Lines 228-229: grammar, see above

Table 1: please use the same font size for all response variables (glucosinolates are larger than the others).

Table 2: given the information presented in table 1, this seems more appropriate as a supplemental table, for readers that want to dig more deeply into the results.

Discussion

Line 257: it has long been known...

Line 261: Thus, our data show...

Line 263: floral phenotypes...

Lines 264-266: this is where I felt there was latitude for more specific a priori hypotheses and predictions.

Lines 276-277: See above – I think that it is a mistake to reveal this information only now in the discussion, rather than framing the study from the introduction by stating that limestone is a poorer soil than tuff (for whatever reasons).

Line 282: because the presence of defensive compounds in nectar...

Line 287: Although herbivory in our experiment did not...

Line 293: this sentence is written as a general comment on pollinator attraction, but the value of references 33 and 69 is that they confirm the truth if this statement for the experimental system used here. It IS true that scent and plant height also contribute to pollinator attraction in other systems, but the wording of this sentence should clarify the importance for this experimental system (i.e. validation of the statistical outcomes in the present study).

Line 307: than in allopatry

Line 318: the patterns of divergence documented here...

Line 381: MANOVA

Line 394: please add Snogerup et al. 1990 to the reference list and cite here as a number

Line 401: was analyzed

Line 402: why not sterilize the soil? You are including soil microbiome variables that might vary independently from tuff vs. limestone, simply due to geographic location.

Line 403: this (added fertilizer after generation 5) does complicate things. I’m not sure how this impacts the interpretation of your results.

Line 416: to establish an aphid colony

Line 440: to gain experience

Line 442: non-evolved is awkward. Perhaps “from unselected lines”?

Line 466: can you replace “bottom to top” with more suitable botanical terms?

Line 490: plant traits

Line 519: on the day that bioassays were conducted

Lines 521-522: awkward wording, please revise

Lines 559-572: you are performing a binary choice assay between (putatively) representative plants from different treatments competing against a control (G1) plant that has not experienced any evolutionary treatments. This means that your null expectation is no change in attractiveness, represented by an “attractive index” in which the null response is 0.5 or 50%. One could imagine a horizontal line at 50%, with a control set of trials competing two G1 plants against each other, resulting in a bar at 50%. Different combinations of soil, pollination and herbivory treatments might make flowers more OR less attractive than the G1 control plants. However, that is NOT what is shown in Fig. 1F.. in fact, I do not understand what is being shown in Fig. 1F.

Lines 573-585: the same can be said for aphid choice assays. How does what you describe here translate into a summary graphic? I do not see a panel comparable to Fig. 1F for aphid choices...

Reviewer #2:

Remarks to the Author:

This manuscript reports on an exciting study in which experimental evolution was used to uncover how adaptation to soil, herbivory, and pollination affect plant trait divergence. The authors designed an elegant, fully-factorial experiment, in which eight generations of selection were imposed on plants, and in which phenotypic trait values and divergence among treatments was then quantified. The authors found phenotypic divergence driven by all three conditions, as well as interactive effects among them. Effects of soil and pollination (hand or bumble bee) were strong, while effects of herbivory were weaker. Interestingly, the highest phenotypic divergence was observed between plants that were grown in different soils and that were bumble bee pollinated.

General comments:

The study is novel, represents a considerable amount of effort, and builds nicely on past work by the same group. However, the manuscript would benefit from the inclusion of some key details that would allow for the evaluation of its relevance beyond the study system. First, a clear motivation for why the particular treatments were chosen should be provided in the introductory part of the manuscript (some but not all of this information is included in the methods section). This includes (1) specifying what is different about the soils in terms of their nutrient content, porosity, or other characteristics relevant to plant growth; and (2) clarifying why the hand pollination treatment was included. I assume the hand pollination treatment represents the maximum amount of possible pollen that could be deposited on the stigma? Quantitative data that shows differences in the amount of pollen deposited on stigmas from each treatment, or seeds produced in each generation from the different treatments (at the individual level) would help contextualize the relevance of these two treatments.

Figure 1 shows trait values for each experimental group after selection. It would be helpful for readers to be able to understand what trait values were before selection, to ensure that differences found after selection were not due to prior differences among groups. Similarly, Figure 4 shows overall phenotypic change, and it would be helpful for the authors to provide information about starting conditions (perhaps plants that changed more started from a different place?).

Throughout the results section, it would be helpful for authors to include estimates of how much traits differed, for example, mean time to flower was XXX longer in the XXX group.

Specific comments:

Line 18: It would be helpful for this sentence to be split into two for clarity.

Line 21: Given the wording, it was unclear if (1) the strongest phenotypic differences were found between bumble bee pollinated plants grown in one soil type, and hand pollinated plants grown in the other soil type, or if (2) the strongest divergence was found between bumble bee pollinated plants in one soil type, and bumble bee pollinated plants in the other soil type. From the results section it seems like the latter is true, and this should be clarified.

Line 72: Replace "less well known" with "their being less research"

Line 74: It would be helpful for the authors to include some specifics about the biology that is being referred to here.

Line 88: It would be helpful for the authors to clarify why they made this prediction.

Line 96 and throughout the results section: It should be clarified if these results are relative to generation one, or to other treatments.

Line 121: Please clarify - effect on what?

Line 133: Please clarify – in contrast to what?

Lines 151-156: Please put this paragraph into context by briefly describing both the bee choice experiment and the alate choice experiment prior to providing the results.

Figure 1: Please specify what G1 is in the caption.

Line 159: Please clarify what “and/or” means here

The term “biotic pollination” is a little convoluted, and since there is only one “biotic” treatment I think it would be clearer for the authors to use the terms “bee pollination” and “hand pollination”.

There are grammatical errors that should be corrected throughout the manuscript. Some examples are: line 40 (remove the second “as well”), line 41 (“root”, not “roots”), line 74 (“suggesting”), line 434 (“used only”), and line 521 (“leaf”). There a number of other corrections that should be made.

Figure 3 shows only traits that showed significant changes, and statistics were done on only these traits. This rationale for using only traits that show significant changes should be provided.

Line 433: The “unpublished data” should be provided.

Line 451: The authors should discuss why were only seven bees were allowed to visit plants and why they were captured after visiting only five plants. Was this intentional, to impose pollen limitation at the replicate level? A description of levels in natural populations (in addition to the citation) would be helpful.

Line 518: If this was not always the bioassay date, using “bioassay date” in the formula would be clearer.

Line 533: The authors should specify if this means exactly two hours.

Table 1: It would be helpful for the authors to find a way to present the data pertaining to interactions in a way that allows the reader to interpret them. Perhaps using the package sjPlot to provide graphs in the supplementary material?

Tables 1&2: If space is a concern, it seems to me these could go in the supplementary material, and additional information that is currently in the methods could be added to the manuscript (please see general comments for additional information that would be helpful).

We would like to thank both reviewers for their constructive comments; we appreciate their efforts in improving the paper.

Reviewer #1 (Remarks to the Author):

This manuscript builds upon a series of recent, high impact experimental evolution studies by Schiestl and associates, using rapid cycling Brassica rapa plants as a model system to track the factorial effects of pollinators, herbivores, and their various combinations on trait evolution over a surprisingly short period of selection (8-10 generations). These previous studies were important not only for their surprising results, but also for the authors' unbiased analysis of trait combinations, openness to unexpected outcomes and synergistic effects, and the generation of novel ideas for future research that followed logically from their findings.

Thus, I was excited to read the present manuscript, which builds logically on these foundations by adding soil type (both for direct and indirect effects) as an experimental variable. There are good reasons to expect the "soil-pollinator" effect detailed by the authors here, going back to early work by Verne Grant on "pollinator climate" and subsequent critics of Grant's work on the Aquilegia system. Edaphic specialization is a major theme in plant diversification, and decoupling soil and pollinators in an experimental design is valuable and overdue.

I would like to separate my remarks into two categories.

1) Effort and impact: the current study summarizes a remarkable amount of work and reveals (I think) an important set of non-random interactions between soil, bee pollination and history of herbivory. Anything more than a cursory glance would reveal the enormous effort invested into a) generational replication of experimental conditions, including hand-pollination and controlled bee- and aphid exposure, b) the large number of standardized measurements, including different kinds of chromatographic separation, and c) highly replicated behavioral assays with bees and aphids. Considering all the work summarized here, much of it undoubtedly performed during COVID-19 lockdowns, it is remarkable that the study has only two authors. The summary figures are so derivative that it is a bit difficult to interpret them at first glance, but if the authors' interpretations are correct, they have identified a potentially fascinating link between biotic and abiotic drivers of plant diversification in a controlled lab experiment. However, I have some concerns about how to interpret soil differences, when the authors added fertilizer after Generation 5 and decided not to autoclave or sterilize the soil before Generation 1. "Soil differences" in this study appear to mean many things... not all of which were controlled. Thus upon closer examination, the study may not be as elegantly designed as one might hope (e.g. beginning with common soil and adding or withholding key macro or micronutrients, microbes, etc.).

Thanks for these comments. We agree that although soil composition was analyzed and the differences between soil types were reported in an earlier study (Dorey and Schiestl 2022), there remain some unknowns about the soil differences, such as physical-chemical parameters and microbial composition. In experiments like ours, one has to find a good balance between natural conditions, and controlled factors. The more natural the conditions are, the more they represent a real natural setting, but the less controlled the experiment is, making interpretations harder. Sterilizing soil could have been done, in order to make the system more controlled, albeit less natural. We do strongly believe, however, that sterilizing soil would not have been a good idea for the experiment. First, it is virtually impossible to keep sterilized soil sterile throughout a growing season, not to think of between generations. Then, with sterilized soil, one can get weird re-settlement patterns, for example some fungi suddenly proliferating explosively. Thus, trying to make the soil environment more controlled, but less natural by sterilizing it, may lead to just unnatural condition, but not necessarily to a better controlled environment. One more alternative could have been to use sand, with artificial nutrients applied; This would have been OK, but simply led to less natural soil

conditions in the experiment. Because we were not interested in detailed aspects of how soil nutrients affect pollinator mediated selection (this may be interesting for follow up studies), we feel that the choice for more natural setting, albeit with soil as a black box is justified. Adding nutrients after generation 5 admittedly made the soil conditions less natural, but this was simply necessary to allow the plants to produce enough seeds for the continuation of the experiment. We do explain this in the text and also show that the addition of nutrients did not alter differences in nutrient limitation between soil types.

2) Outline and presentation: despite my interest in the findings of this study, I found many problems in the current outlay of this manuscript. First, it simply does not work well in the Nature format. I provide numerous examples below of places where the reader does not have sufficient information to understand the motivation or justification for the study, or even how the soil types differed, before the results are presented and interpreted. Some of this is revealed afterwards in the Methods, but by then it is too late. And while I like the idea of replicating some of the natural conditions for *Brassica incana* and its aphid antagonists in southern Italy (rather than continuing to build on the previous studies using *Pieris* larvae as herbivores), this information is lacking in the Introduction, and Figure 5 comes far too late for the reader to appreciate the experimental design. Second, I was disappointed that the authors did not develop or test a priori hypotheses for nutrient tradeoffs, which were only mentioned later in the Discussion. There is a large body of theory that could have guided such hypotheses, especially given the life history parameters of rapid cycling *B. rapa*. Thus, the Introduction comes across as unnecessarily cursory and exploratory. Third, some of the response variables are summarized in highly derived ways, and I wonder about some basic aspects, such as a) how did the authors deal with collinearity of chemical response variables and correlated morphological traits, and b) how did they handle variance measures for individual variables when combining them to create aggregate variables (e.g. for all volatiles or all morphological traits)? I do not understand how the bee and aphid behavioral assays described in the methods translated into results analyzed statistically or summarized in figures/tables. The figures don't show this very clearly, in my opinion. Finally, the manuscript is full of small, distracting editorial mistakes that should have been caught and corrected before submission. It is a level of sloppiness unsuitable to this (or any) journal.

Below, I provide specific comments and questions.

Introduction: could have used another round of proof-reading, given the number of editorial errors:

Line 36: neutral, not neural. Done L37

Line 57: pollinators, in contrast, act...done L59

Line 72: although its effects on floral evolution are less well known, water availability...done L74

Lines 74-75: awkward wording, please reword this passage. Done L79

Lines 80-81, Figure 5: This passage is key to setting up your experimental design and expectations for readers but the backwards format of Nature makes it feel like an afterthought here. Figure 5 really should be presented as Figure 1, with more explanation and justification for the reader. Similarly, the passage from lines 390-400 creates a very different and more specific picture for the reader: the authors are intending to replicate natural conditions (e.g. for *Brassica incana*) in Italy, in which limestone vs. tuff soils, bee pollination and aphid herbivory are key ecological variables. At this point in the introduction, the reader has no idea of this. For example, having read previous papers by this group, one might expect the use of *Pieris* butterfly larvae, rather than aphids, as the herbivore treatment. How do limestone and tuff soils differ? What should we expect? Later, the authors mention having performed a component analysis of the soils, but those results are not presented.

We have now worked out the last paragraph of the introduction to include more of this information, i.e. the design and the rationale for the design. We also place the former Figure 5 here as Figure 1

Line 81: soil, pollination and herbivory text has been removed

Line 87: and if so, which combinations lead to... text has been removed

Lines 88-90: “Initially, we predicted that more factors should cause more divergence, and thus the interaction of both herbivory and pollination lead to the strongest divergence among plant populations on different soil types” done. This section was re-organized. L108

This is a weak closing sentence to a generally strong introduction. Did the authors not have more specific hypotheses and predictions to test? I appreciate the exploratory nature of this approach, but I was expecting to see a stronger a priori conceptual or theoretical component framing alternative outcomes here. For example, if soil N is limiting, would that constrain responses to selection for floral display (flower size, number) or scent (e.g. lines 102-103), and would this be exacerbated when under attack by herbivores (thus increasing demand for N-related defensive chemistry)? Again, as above, the reader does not yet know what the soil treatments are. One could imagine starting with a generic potting soil and manipulating micronutrients, particle size or pH, in a controlled fashion.

We have re-organized this section, putting more focus on predictions for the research questions.

Results:

Lines 96-97: “plants that evolved in tuff” – I understand what the authors are saying here, but it is confusing to the reader given the possibility that they are comparing plants whose seeds were collected in locations with tuff vs. limestone soil. Instead, they could write “plants that experienced tuff as a soil treatment...” We added “during the experiment” to make clear the past evolution refers to our experiment L133

Line 120: revise with punctuation Done. Paragraph was recrafted.

Line 143: the factor “pollination” done L193

Line 140-147: how many of these traits were correlated? We include now a correlation heatmap to show the correlations between traits (Figure S2)

Line 160: first mention of “haldanes” as a unit: please define in parentheses. done

Line 162: in categories, with morphological.... This should be “among”, now in L215

Figure 1: the icons above and within each bar are too small to be appreciated when viewing this figure. Vertical axes are lacking units in panels C, D, E and F. Please provide them directly or in the figure legend. Please explain what G1 means in the figure legend. Perhaps also mention what was done to accommodate multiple comparisons in this figure. Corrected (Figure 1 is now Figure 2).

Figure 2: please explain briefly in the figure legend that these are aggregate traits. How are the error bars reflective (or not?) of variance across the combined traits? Done. L242-243 (Figure 2 is now Figure 3)

Figure 3 legend: lines 204-205, grammar: please check treatments and groups (singular vs. plural). Please add one sentence to the legend helping to interpret the results summarized herein, rather than simply referring the reader to Table S4. The limestone and tuff-treated lines segregate left to right (DF1) in the bee pollination panel and to a lesser extent in the hand pollination panel, but not top to bottom (DF2) in either panel. How might you concisely summarize this outcome?

done – an interpretation sentence was added. (Figure 3 is now Figure 4).

Figure 4, panel C especially: one might conclude that “limestone, hand pollination, no herbivory” leads to the same outcome as “tuff, bee pollination and herbivory” for mean evolutionary rates regarding floral scent, but this is due to selection for different scent compounds (lines 130-136).

done - a cautionary note was added to the legend. L275 (note that Figure 4 became Figure 5).

Lines 228-229: grammar, see above corrected

Table 1: please use the same font size for all response variables (glucosinolates are larger than the

others). corrected

Table 2: given the information presented in table 1, this seems more appropriate as a supplemental table, for readers that want to dig more deeply into the results. done

Discussion

Line 257: it has long been known done L307

Line 261: Thus, our data show done L311

Line 263: floral phenotypes done L313

Lines 264-266: this is where I felt there was latitude for more specific a priori hypotheses and predictions. More was added L314-319

Lines 276-277: See above – I think that it is a mistake to reveal this information only now in the discussion, rather than framing the study from the introduction by stating that limestone is a poorer soil than tuff (for whatever reasons). We added the information about different nutrient availability to the paragraph at the end of the introduction

Line 282: because the presence of defensive compounds in nectar done L335

Line 287: Although herbivory in our experiment did not done L340

Line 293: this sentence is written as a general comment on pollinator attraction, but the value of references 33 and 69 is that they confirm the truth if this statement for the experimental system used here. It IS true that scent and plant height also contribute to pollinator attraction in other systems, but the wording of this sentence should clarify the importance for this experimental system (i.e. validation of the statistical outcomes in the present study).

OK, a sentence was added to make this more clear L346-347

Line 307: than in allopatry done L360

Line 318: the patterns of divergence documented here done L371

Line 381: MANOVA done L441

Line 394: please add Snogerup et al. 1990 to the reference list and cite here as a number done

Line 401: was analyzed done L456

Line 402: why not sterilize the soil? You are including soil microbiome variables that might vary independently from tuff vs. limestone, simply due to geographic location.

We explain this in the first response to the reviewer, as well as now in the last paragraph of the introduction.

Line 403: this (added fertilizer after generation 5) does complicate things. I'm not sure how this impacts the interpretation of your results.

We explain the rationale for this and why it unlikely impacted our results in the first comment above

Line 416: to establish an aphid colony done L471

Line 440: to gain experience done L494

Line 442: non-evolved is awkward. Perhaps "from unselected lines"? done L497

Line 466: can you replace "bottom to top" with more suitable botanical terms? done L522

Line 490: plant traits done L546

Line 519: on the day that bioassays were conducted changes L575; formula was changed in respect of reviewer 2's comment.

Lines 521-522: awkward wording, please revise done L576

Lines 559-572: you are performing a binary choice assay between (putatively) representative plants from different treatments competing against a control (G1) plant that has not experienced any evolutionary treatments. This means that your null expectation is no change in attractiveness, represented by an "attractive index" in which the null response is 0.5 or 50%. One could imagine a horizontal line at 50%, with a control set of trials competing two G1 plants against each other, resulting in a bar at 50%. Different combinations of soil, pollination and herbivory treatments might

make flowers more OR less attractive than the G1 control plants. However, that is NOT what is shown in Fig. 1F... in fact, I do not understand what is being shown in Fig. 1F.

There is a misunderstanding in terms of how we performed the bioassays. We did not perform binary choice assays with two plants in each bioassay run, but tested always one plant of all treatment groups, plus one of generation one together (i.e. 9 plants were tested together) using three bees for each run. This was done 71 times, leading to a total of 639 plants assayed with 213 bees. This now described in a clearer way (L617).

Lines 573-585: the same can be said for aphid choice assays. How does what you describe here translate into a summary graphic? I do not see a panel comparable to Fig. 1F for aphid choices...

The aphid choice assays were done in the same way as the bee assays, as is now clearly explained (L617, L631). The results of these tests were non-significant, thus no figure was included.

Reviewer #2 (Remarks to the Author):

This manuscript reports on an exciting study in which experimental evolution was used to uncover how adaptation to soil, herbivory, and pollination affect plant trait divergence. The authors designed an elegant, fully-factorial experiment, in which eight generations of selection were imposed on plants, and in which phenotypic trait values and divergence among treatments was then quantified. The authors found phenotypic divergence driven by all three conditions, as well as interactive effects among them. Effects of soil and pollination (hand or bumble bee) were strong, while effects of herbivory were weaker. Interestingly, the highest phenotypic divergence was observed between plants that were grown in different soils and that were bumble bee pollinated.

General comments:

The study is novel, represents a considerable amount of effort, and builds nicely on past work by the same group. However, the manuscript would benefit from the inclusion of some key details that would allow for the evaluation of its relevance beyond the study system. First, a clear motivation for why the particular treatments were chosen should be provided in the introductory part of the manuscript (some but not all of this information is included in the methods section). This includes (1) specifying what is different about the soils in terms of their nutrient content, porosity, or other characteristics relevant to plant growth;

We have added now a table showing the chemo-physical properties of each soil type (Table S10) and (2) clarifying why the hand pollination treatment was included. I assume the hand pollination treatment represents the maximum amount of possible pollen that could be deposited on the stigma? Quantitative data that shows differences in the amount of pollen deposited on stigmas from each treatment, or seeds produced in each generation from the different treatments (at the individual level) would help contextualize the relevance of these two treatments.

Hand pollination was included as a control treatment, as it represents efficient pollination, but without selection. Bumblebees also pollinate highly efficiently, but they select specific plants according to their preferences. We have added a sentence making this more clear in the newly crafted last paragraph of the introduction L94.

Figure 1 shows trait values for each experimental group after selection. It would be helpful for readers to be able to understand what trait values were before selection, to ensure that differences found after selection were not due to prior differences among groups. Similarly, Figure 4 shows overall phenotypic change, and it would be helpful for the authors to provide information about starting conditions (perhaps plants that changed more started from a different place?).

The starting values are in fact given in the figure, labeled as G1. We explain now in the legend that G1 means plants of generation 1, and the values are those before selection (L230). The values in Figure 4 (now Figure 5) are calculated from values of plants of generation 1 and 10, and no values of generation 1 can be shown.

Throughout the results section, it would be helpful for authors to include estimates of how much traits differed, for example, mean time to flower was XXX longer in the XXX group. Text was adapted to include estimates of how much traits differed. L133 to L186

Specific comments:

Line 18: It would be helpful for this sentence to be split into two for clarity. Done L20

Line 21: Given the wording, it was unclear if (1) the strongest phenotypic differences were found between bumble bee pollinated plants grown in one soil type, and hand pollinated plants grown in the other soil type, or if (2) the strongest divergence was found between bumble bee pollinated plants in one soil type, and bumble bee pollinated plants in the other soil type. From the results section it seems like the latter is true, and this should be clarified.

OK, this was changed to: "We found the strongest evolutionary divergence when plants evolved in different soils, with bee-pollination rather than hand-pollination, irrespective of herbivory." L23

Line 72: Replace "less well known" with "their being less research" changed to "less well investigated" L74

Line 74: It would be helpful for the authors to include some specifics about the biology that is being referred to here. Done L81

Line 88: It would be helpful for the authors to clarify why they made this prediction. More was added here L106.

Line 96 and throughout the results section: It should be clarified if these results are relative to generation one, or to other treatments.

This is indicated in the heading: L130 and 209 (i.e. "measured between treatment groups of generation 10) or (measured between between generation 1 and 10")

Line 121: Please clarify - effect on what? Deleted. Paragraph was fully changed to include estimates of traits changes.

Line 133: Please clarify – in contrast to what? Deleted. Paragraph was fully changed to include estimates of traits changes.

Lines 151-156: Please put this paragraph into context by briefly describing both the bee choice experiment and the alate choice experiment prior to providing the results. A transition sentence was added: L201

Figure 1: Please specify what G1 is in the caption. Done L230

Line 159: Please clarify what "and/or" means here done L212

The term "biotic pollination" is a little convoluted, and since there is only one "biotic" treatment I think it would be clearer for the authors to use the terms "bee pollination" and "hand pollination". We changed "biotic" to "bee" pollination when it referred to our experiment (e.g. L357), and left "biotic pollination" when it referred generally to one type of pollination (L27). The title was also change to "bee-pollination".

There are grammatical errors that should be corrected throughout the manuscript. Some examples

are: line 40 (remove the second “as well”) done L40, line 41 (“root”, not “roots”) done L41, line 74 (“suggesting”) done L77, line 434 (“used only”) done L489, and line 521 (“leaf”) done L577. There a number of other corrections that should be made.

Figure 3 shows only traits that showed significant changes, and statistics were done on only these traits. This rationale for using only traits that show significant changes should be provided. done L255 (now Figure 4).

Line 433: The “unpublished data” should be provided.

This is an unpublished PhD thesis that is not online available; we feel this data is not of key importance because many Brassica species are pollinated by bumblebees, and we cite two references that document this.

Line 451: The authors should discuss why were only seven bees were allowed to visit plants and why they were captured after visiting only five plants. Was this intentional, to impose pollen limitation at the replicate level? A description of levels in natural populations (in addition to the citation) would be helpful. Information provided in line 510-512

Line 518: If this was not always the bioassay date, using “bioassay date” in the formula would be clearer. Done L571: “(30 + 1 – time to flower), where 30 represents the age of 519 the plants at bioassays date” was changed to “flower duration, where flowering duration represents the number of days a plant had flowered before bioassays.”

Line 533: The authors should specify if this means exactly two hours. Done L589

Table 1: It would be helpful for the authors to find a way to present the data pertaining to interactions in a way that allows the reader to interpret them. Perhaps using the package sjPlot to provide graphs in the supplementary material?

We chose not to do this, because we have so many interactions and putting them in a graph would make the results pretty messy and confusing for the readers.

Tables 1&2: If space is a concern, it seems to me these could go in the supplementary material, and additional information that is currently in the methods could be added to the manuscript (please see general comments for additional information that would be helpful).

Table 2 was moved to the supplementary

Dorey, T., and F. P. Schiestl. 2022. Plant phenotypic plasticity changes pollinator-mediated selection. *Evolution* **76**:2930-2944.

Reviewers' Comments:

Reviewer #1:

Remarks to the Author:

The authors received a substantial list of editorial concerns from two reviewers and have submitted a revised manuscript attempting to address many of these concerns.

One major problem evident in the first submission was that basic problems in the way the paper was written (organization, syntax, grammar) made it difficult to fully digest the complex data sets and statistical treatments resulting from the authors' extensive experimental design. Unfortunately, that situation has not improved in the revised text. Many of the topic sentences for new paragraphs were laden with stylistic or grammatical problems or did not effectively convey the importance of the material being introduced to the wider readership of this journal. I list some examples of this below, but was not willing to address each error (e.g. subject-verb agreement, personification) throughout the manuscript. This was deeply frustrating, given the extraordinary data set presented by the authors, but it remains very difficult to read and digest this study.

Another issue carrying over from the initial manuscript was the need to outline and visualize the experimental design more clearly for the reader before the presentation of results. The new "Figure 1" provides some improvement in this area, although an additional flow chart would be even more effective.

Finally, a criticism of the original manuscript was that the Introduction did little to outline the conceptual framework in which this study is best considered. Instead, the authors used the Discussion to "retro-fit" some of their more intriguing results to various theories concerning tradeoffs in life history parameters and specific traits when plant defense and reproductive success are limited by soil nutrients.

Unfortunately, the additional text provided by the authors does little to address this problem. The reader is left with the impression that this study really was conceived and performed in an exploratory way, with the expectation that something interesting would result, given the success of previous "experimental evolution" studies on this system. And something interesting DID emerge, but given the many differences between soil types here, there could be many reasons for those differences, leaving the reader with tantalizing phenomena that (as yet) lack strong inference.

Abstract

Line 13: "and is a main motor..." is a bit awkward

Line 15: "adaptation... is thought"

Line 21: "plants that had experienced aphid herbivory" rather than "with an evolutionary history of herbivory".

Line 25: "in diverse suites of traits" rather than "in all kinds of traits", the latter is a bit informal

Introduction

The first sentence is not sufficiently impactful for a Nature Communications paper.

Line 43: divergence... is expected to be selected...

Line 46: and is well-documented

Line 58: strong selection on defense... mechanisms? Strategies? Phenotypes? Missing something here

Line 60: morphological traits

Lines 65-67: wording and syntax are awkward: floral traits are attractive but floral rewards usually are not the attractants.

Line 73: compensate for or tolerate

Lines 63-76: most of this content is correct but perhaps could be explained more elegantly regarding energetic vs. ecological costs and resource limitation.

Line 77: "Despite theory and experimental work suggest that multiple factors impact plant evolution"

Awkward syntax, please revise. Through line 80, lots of personification (e.g. studies don't attempt or make inferences, scientists do).

Line 87: on limestone OR tuff soil, not both at the same time

Line 103: what is meant here by "defense"? A specific defensive trait, a response variable measured in aphid fitness, or a response variable measured in plant fitness?

Line 108: what is the rationale for "more factors causing more divergence", when it is possible that more factors might cancel each other out, leading to a net zero divergence.

Neither of these questions or their attendant predictions are rooted in (potentially competing) bodies of theory. It is clear from this section that the study really is exploratory and descriptive, rather than addressing specific conceptual gaps or conflicting hypotheses emerging from the post-Strauss/Armbruster literature on complex selective forces acting on floral evolution and diversification. I am not denigrating the experiment itself, but the authors have missed an opportunity to frame their study in a more scholarly way.

Figure 1: I like this figure very much where it is currently placed, in the beginning of the manuscript. Please clarify in the figure legend what (if any) is the difference between replicates having a pale green background vs. a pale blue background. In all remaining figures, green is used to indicate tuff and blue is used for limestone. That should be stated somewhere.

Results

Line 150: I suggest writing "aphid herbivory" in parallel with "bee pollination", just to remind readers that you are not tracking more typical modes of leaf-chewing herbivory (e.g. by larval Lepidoptera).

Line 202: incorrect use of evidenced

Table 2: P values should have decimals, not commas. Ditto for Table S1

Discussion:

Lines 330-332: this is a limitation in terms of fitting the experimental results into a conceptual space: aphids are not the ideal herbivore for tracking the effects of glucosinolates

Lines 343-345: please restructure this sentence, the syntax is problematic

Reviewer #2:

Remarks to the Author:

I reviewed this manuscript before and the clarity has improved. I appreciate that quantitative differences among treatments in phenotypes are now presented. However, some statistics seem to be lacking still (please see below). More generally, the manuscript would still benefit from the readers being introduced to what was measured prior to reading the results section. The way that statistical results were obtained should be specified at least in general terms before results are presented, and it should be clear if the differences outlined in the results section represent significant effects or not. It would be helpful if in the introduction the authors explained what chemical analyses were performed and what was collected for analysis. As it reads, the results detailing changes in specific compounds could be better contextualized. The relevance of dividing results into scent vs glucosinolates, in particular, should be clarified for a broad audience. What standardized soil is should be clarified.

Line 16: Replace "between" with "among"

Line 21: The semi-colon is confusing and it would be clearer to specify that the effect was dampened by herbivory.

Line 73: Compensate for

Line 82: Grown in

Line 83: and by performing

Line 85: This sentence would benefit from additional clarity/being split into two

Line 88: This seems out of place, perhaps add this as motivation earlier and introduce the term model system at that point (rather than line 112)

Line 108: More factors causing more divergence seems generalized, and it would be helpful for the authors to include a rationale based on past work as suggested by Reviewer #1.

Line 122: study, in which 98 full-sib

Line 125: This would benefit from clarity. Perhaps "Soil types are indicated by different colors with brown representing XXX and grey representing XXX"

Line 130: The part in parentheses should be put earlier and it would be clearer for it to be removed here

Line 135: Should this be "Tables"?

Line 204: Specify how this was done briefly

Line 150: It would be helpful for the authors to present the results such that numbers above 1 indicated bigger relative to controls and numbers less than 1 indicate smaller, here and throughout.

Line 180: The way this makes it written makes it seem like the results might be cherry picked, and that this paragraph may not comprehensively reflect the results. Were any lower? The numbers should also be linked to the compound unless all increased, rather than stating the range for "some" compounds.

Line 195: Add which were not significant as well

Line 207: Remove "in contrast"

Figures 2 & 4: What bold indicates needs to be specified

Line 309: generations

Line 366: trajectories

Line 376: shifts

Line 380: a stronger last sentence would benefit the manuscript, or "even more" should be clarified.

Line 409: Remove "a"

Line 418: families

Line 418: were randomly selected for use in the experimental evolution experiment

Line 437: provide the numbers to back up this claim

Line 454: this is a better place for the discussion about soil sterilization (rather than prior to the methods – it seemed out of place there)

Line 464: exposed either

Line 477: levels

Line 659: This paragraph and the following one have numerous grammatical errors that should be fixed

Figure S1: What each bar means and what each letter means needs to be specified in the caption.

Line 986: for plant traits

We have now one more time revised our MS and feel the manuscript has significantly improved. Below, we explain in detail how we have addressed the reviewers' concerns. We would like thank the reviewers for the hard work they have invested in helping to improve the manuscript!

Reviewer #1

The authors received a substantial list of editorial concerns from two reviewers and have submitted a revised manuscript attempting to address many of these concerns.

One major problem evident in the first submission was that basic problems in the way the paper was written (organization, syntax, grammar) made it difficult to fully digest the complex data sets and statistical treatments resulting from the authors' extensive experimental design. Unfortunately, that situation has not improved in the revised text. Many of the topic sentences for new paragraphs were laden with stylistic or grammatical problems or did not effectively convey the importance of the material being introduced to the wider readership of this journal. I list some examples of this below, but was not willing to address each error (e.g. subject-verb agreement, personification) throughout the manuscript. This was deeply frustrating, given the extraordinary data set presented by the authors, but it remains very difficult to read and digest this study.

We have continued to try our best to improve the text.

Another issue carrying over from the initial manuscript was the need to outline and visualize the experimental design more clearly for the reader before the presentation of results. The new "Figure 1" provides some improvement in this area, although an additional flow chart would be even more effective.

Finally, a criticism of the original manuscript was that the Introduction did little to outline the conceptual framework in which this study is best considered. Instead, the authors used the Discussion to "retro-fit" some of their more intriguing results to various theories concerning tradeoffs in life history parameters and specific traits when plant defense and reproductive success are limited by soil nutrients.

Unfortunately, the additional text provided by the authors does little to address this problem. The reader is left with the impression that this study really was conceived and performed in an exploratory way, with the expectation that something interesting would result, given the success of previous "experimental evolution" studies on this system. And something interesting DID emerge, but given the many differences between soil types here, there could be many reasons for those differences, leaving the reader with tantalizing phenomena that (as yet) lack strong inference.

We have now re-worked the introduction and the discussion part with a focus on developing more the conceptual framework this study, namely plant divergent evolution and diversification. We cite several studies more that have found effects of soil types on flower traits and pollinators, interactive effects of herbivory and pollination, and macroevolutionary diversification in response to soil and pollinator types. We use these as a framework to argue for the need for strictly experimental studies that highlight the importance of individuals ecological factors and their interaction.

Abstract

Line 13: "and is a main motor..." is a bit awkward

L13: Changed to "promotes diversification"

Line 15: "adaptation... is thought"

Done (L15)

Line 21: “plants that had experienced aphid herbivory” rather than “with an evolutionary history of herbivory”.

Done (L23)

Line 25: “in diverse suites of traits” rather than “in all kinds of traits”, the latter is a bit informal

Done (L27)

Introduction

The first sentence is not sufficiently impactful for a Nature Communications paper.

Changed now (L38-41)

Line 43: divergence... is expected to be selected...

Done (L46)

Line 46: and is well-documented

Done (L49)

Line 58: strong selection on defense... mechanisms? Strategies? Phenotypes? Missing something here

Added L61: “mechanisms in plants”

Line 60: morphological traits

Done (L64)

Lines 65-67: wording and syntax are awkward: floral traits are attractive but floral rewards usually are not the attractants.

This paragraph (starting L67) has been completely re-worked.

Line 73: compensate for or tolerate

Done (L87 and 90)

Lines 63-76: most of this content is correct but perhaps could be explained more elegantly regarding energetic vs. ecological costs and resource limitation.

This part was changed: L80-90

Line 77: “Despite theory and experimental work suggest that multiple factors impact plant evolution”

Awkward syntax, please revise. Through line 80, lots of personification (e.g. studies don’t attempt or make inferences, scientists do).

This section was revised and was incorporated in a new paragraph: L98-111

Line 87: on limestone OR tuff soil, not both at the same time

Done (L120)

Line 103: what is meant here by “defense”? A specific defensive trait, a response variable measured in aphid fitness, or a response variable measured in plant fitness?

Added “trait” and “to pollinators” (L142-143)

Line 108: what is the rationale for “more factors causing more divergence”, when it is possible that more factors might cancel each other out, leading to a net zero divergence.

Added: “because the ecological factors likely selected for different traits” (L148-149)

Neither of these questions or their attendant predictions are rooted in (potentially competing) bodies of theory. It is clear from this section that the study really is exploratory and descriptive, rather than addressing specific conceptual gaps or conflicting hypotheses emerging from the post-Strauss/Armbruster literature on complex selective forces acting on floral evolution and diversification. I am not denigrating the experiment itself, but the authors have missed an opportunity to frame their study in a more scholarly way.

We have modified this section so it becomes more clear what the initial motivation for the experiments are, and outline more clearly existing hypothesis/gaps in our knowledge. We have also followed this up in the discussion section, where concepts of diversification are now more clearly explained and discussed in the context of our study.

Figure 1: I like this figure very much where it is currently placed, in the beginning of the manuscript. Please clarify in the figure legend what (if any) is the difference between replicates having a pale green background vs. a pale blue background. In all remaining figures, green is used to indicate tuff and blue is used for limestone. That should be stated somewhere.

Done (L167-168)

Results

Line 150: I suggest writing “aphid herbivory” in parallel with “bee pollination”, just to remind readers that you are not tracking more typical modes of leaf-chewing herbivory (e.g. by larval Lepidoptera).

Done throughout the MS

Line 202: incorrect use of evidenced

Replaced by “documented” (L243)

Table 2: P values should have decimals, not commas. Ditto for Table S1

Corrected

Discussion:

Lines 330-332: this is a limitation in terms of fitting the experimental results into a conceptual space: aphids are not the ideal herbivore for tracking the effects of glucosinolates

Lines 343-345: please restructure this sentence, the syntax is problematic

Done (L384)

Reviewer #2 (Remarks to the Author):

I reviewed this manuscript before and the clarity has improved. I appreciate that quantitative differences among treatments in phenotypes are now presented. However, some statistics seem to be lacking still (please see below). More generally, the manuscript would still benefit from the readers being introduced to what was measured prior to reading the results section. The way that statistical results were obtained should be specified at least in general terms before results are presented, and it should be clear if the differences outlined in the results section represent significant effects or not.

Some of these issues stem from the fact that because of journal style, the results are presented before the methods. We added now that morphological and chemical traits were measured and how they were analyzed (L135). We introduce some statistical analyzes at L140. We do follow the general convention that results that are mentioned are based on statistically significant differences, without mentioning “significant” every time.

To deal with the issue of correlated traits in the analyses, we have now calculated principle components using all traits measured, and re-calculated the discriminant function analysis (DFA) using principle components (PCs; L721). As principle components are orthogonal (uncorrelated), this removes the problem of correlations for the multivariate comparison with DFA. For comparisons based on multiple, individual variables (i.e. the F-value and haldanes analysis), we have not used PCs, because we strive to do this comparison separately for the three different trait categories (morphological, volatiles, and glucosinolates), and the use of PCs would have made this separation hardly possible (because PCs have mixed factor loadings from different variables, see in the new Table S10). Thus, we removed 7 variables that showed the highest correlations with other variables and re-did the F-values and haldanes analysis using the so reduced set of traits (see L708 and L757).

It would be helpful if in the introduction the authors explained what chemical analyses were performed and what was collected for analysis. As it reads, the results detailing changes in specific compounds could be better contextualized.

Done (L135)

The relevance of dividing results into scent vs glucosinolates, in particular, should be clarified for a broad audience. What standardized soil is should be clarified.

Done (L137); what standardized soil means was precised in (L471)

Line 16: Replace “between” with “among”

Replace by “effects of multiples ecological factors” (L17)

Line 21: The semi-colon is confusing and it would be clearer to specify that the effect was dampened by herbivory.

The sentence was divided in multiple sentences (L21-23)

Line 73: Compensate for

Done (L87 and L90)

Line 82: Grown in

This part was replaced (L114)

Line 83: and by performing

This part was replaced in the paragraph (L112)

Line 85: This sentence would benefit from additional clarity/being split into two

Part of the sentence was deleted (L114)

Line 88: This seems out of place, perhaps add this as motivation earlier and introduce the term model system at that point (rather than line 112)

I realize that this part is a little odd here, but we were asked to explain why we set the experimental parameters as they were, here, and this is what this section is for.

Line 108: More factors causing more divergence seems generalized, and it would be helpful for the authors to include a rationale based on past work as suggested by Reviewer #1.

Done (L148)

Line 122: study, in which 98 full-sib

Done (L164)

Line 125: This would benefit from clarity. Perhaps "Soil types are indicated by different colors with brown representing XXX and grey representing XXX"

An explanation of the different colored squares (blue or green, representing the two soil types used in this study) was added as suggested by reviewer one. This information would help the reader in understanding what the two different colors refer to in the next figures. (L164)

Line 130: The part in parentheses should be put earlier and it would be clearer for it to be removed here

We decided to remove the parentheses as we essentially compared treatments lines at generation 10. (L173 and L252)

Line 135: Should this be "Tables"?

Corrected throughout the MS

Line 204: Specify how this was done briefly

Done (L245-246)

Line 150: It would be helpful for the authors to present the results such that numbers above 1 indicated bigger relative to controls and numbers less than 1 indicate smaller, here and throughout.

Indicators were changed as suggested. Following these changes, paragraph one of results was fully rewritten in a clearer way. (L176 onwards)

Line 180: The way this makes it written makes it seem like the results might be cherry picked, and that this paragraph may not comprehensively reflect the results. Were any lower? The numbers should also be linked to the compound unless all increased, rather than stating the range for "some" compounds.

Results were changed according to the previous comment, making distinctions between increases and decreases clearer. We added methyl salicylate, which was previously omitted. In this section, we only indicated traits that showed significant SxPxH in Table S6 as well as a significant PxH effect in Table S9.

We deleted “produced 1.13 to 2.96 times higher amounts of some aromatic compounds” for clarity (L220-221).

Line 195: Add which were not significant as well

We now performed a principal component analysis using all measured traits. Scores of the 12 retained PCs were used to perform the DFA. (PCs and variables loadings were added in Table S10).

Line 207: Remove “in contrast”

Done (L249)

Figures 2 & 4: What bold indicates needs to be specified

Corrected

Line 309: generations

Done (L358)

Line 366: trajectories

Done (L420)

Line 376: shifts

Done (L431)

Line 380: a stronger last sentence would benefit the manuscript, or “even more” should be clarified.

The last sentences were further developed (L434-439)

Line 409: Remove “a”

Done (L466)

Line 418: families

Done (L476)

Line 418: were randomly selected for use in the experimental evolution experiment

Done (L476)

Line 437: provide the numbers to back up this claim

Done (L497)

Line 454: this is a better place for the discussion about soil sterilization (rather than prior to the methods – it seemed out of place there)

We placed this in the introduction upon request of one of the reviewers.

Line 464: exposed either

Done (L522)

Line 477: levels

Done (L536)

Line 659: This paragraph and the following one have numerous grammatical errors that should be fixed

Done (L726)

Figure S1: What each bar means and what each letter means needs to be specified in the caption.

Done (L1094-1096)

Line 986: for plant traits

Done (L1120)